# Synthetic aptamer mechanoreceptors enable cell-specific force sensing and temporal control via DNA circuits

Tao Xu [1], Soumya Sethi[1], Christoph Drees [1] & Andreas Walther [1,2] ✉

Cells interpret mechanical cues from their microenvironment with spatio-temporal precision to guide adaptive behaviors. However, engineering synthetic mechanosensing systems with both cell-specificity and programmability remains challenging, especially when targeting ubiquitous classical mechanoreceptors. Here, we introduce an all-DNA mechanosensing platform based on aptamers that transmit force through noncanonical surface receptors. Aptamer–receptor recognition acts as a molecular gate for force transduction, enabling the design of mechanoprobes with cell-type selectivity. These probes interpret diverse mechanical inputs via distinct mechanisms, including actomyosin-driven contractility and membrane ruffling during macropinocytosis. By integrating aptamer mechanoprobes with upstream DNA reaction networks, we achieve reversible and temporally programmable mechanoresponses. This modular, all-nucleic-acid system offers a general framework for constructing tunable mechanotransduction circuits. It expands the design space for synthetic mechanobiology and provides opportunities for autonomous, multi-layered mechanical–biochemical regulation in tissue engineering, morphogenesis, and dynamic cell programming.

Mechanical forces are key regulators of cellular behavior, shaping processes from morphogenesis to disease progression by guiding how cells sense, adapt, and respond to their microenvironment[1–4]. These responses are typically mediated by mechanoreceptors, which transduce physical stimuli into biochemical signals controlling cell fate, tissue structure, and homeostasis. Understanding and customizing the mechanotransduction process is pivotal for advancing mechanotargeting-based therapies, tissue engineering, and regenerative medicine[5,6]. However, orchestrating orthogonal cell-specific processes and implementing autonomous control mechanisms during complex morphogenesis is still challenged by the limited specificity, versatility, and dynamic tunability of the current mechanosensing toolbox[7–9].

Selectivity and differential force sensing at the cellular level are key design principles for engineering synthetic mechanosensing materials. Cells endogenously express a variety of mechanoreceptors, including integrins, cadherins, T cell receptors, and Piezo channels to accommodate diverse mechanical demands[10–13]. To engage with those, integrin-targeting mechanoprobes (MPs) have been most widely explored[14]. Pioneering studies have employed DNA-based MPs with Arg-Gly-Asp (RGD) motifs to measure piconewton (pN) forces transmitted through integrins. Depending on the DNA architecture, such probes can provide either irreversible (historical) readouts via duplex rupture or reversible (real-time) readouts via hairpin unfolding[15]. Recent innovations have even enabled quantification of single-molecule force loading rates[16,17]. However, the broad and often non-specific expression of integrins across cell types undermines terminal specificity[10,18], limiting their applicability in mixed or heterogeneous cell populations. Greater selectivity can be achieved using protein- or antibody-based ligands conjugated to DNA-MPs, which have proven valuable for interrogating T cell signaling and for constructing mechanosensitive cell-cell junctions[12,19]. In addition to sensing forces,

[1]Life-Like Materials and Systems, University of Mainz, Mainz, Germany. [2]Max Planck Institute for Polymer Research, Mainz, Germany. ✉e-mail: andreas.walther@uni-mainz.de

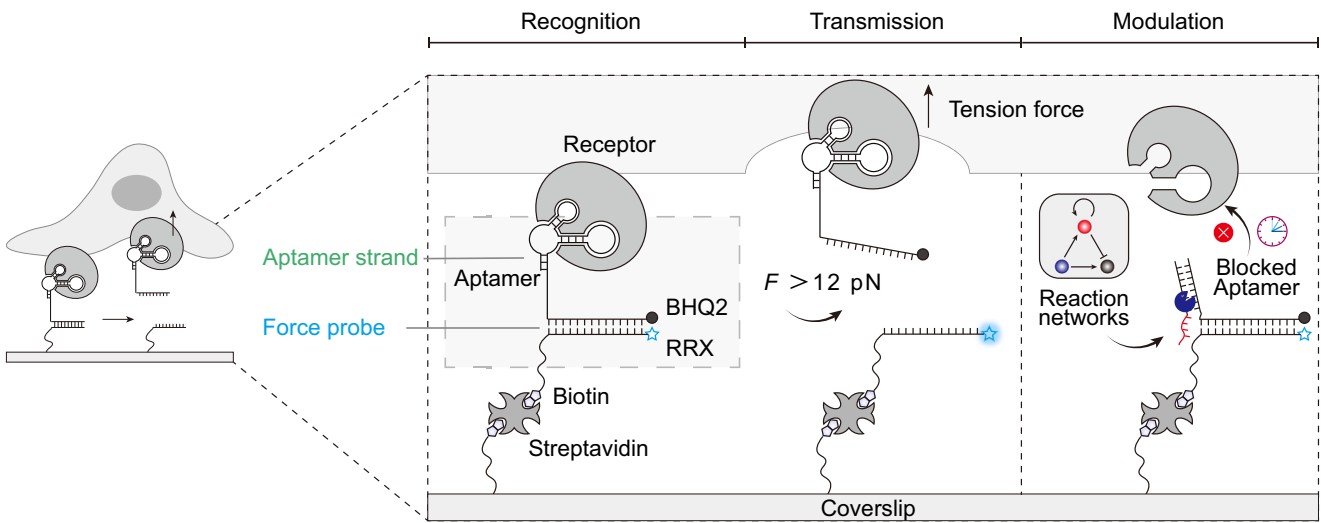

**Fig. 1 | Aptamer-based mechanoprobes (MPs) enable cell-type-specific mechanosensing and programmable control via upstream DNA reaction networks.** The all-nucleic-acid MP consists of a cell-targeting aptamer ligand and a DNA duplex serving as a force sensor. A fluorophore–quencher pair (Rhodamine Red X and BHQ2) transduces mechanical forces into fluorescence signals upon duplex dissociation. Specific force transmission is gated by aptamer–receptor recognition, allowing for cell-type-selective responses. Tuning the geometry of the duplex enables force thresholding to filter defined mechanical inputs. Integration with upstream DNA reaction networks (DRNs) permits programmable and non-linear regulation of aptamer accessibility, establishing a versatile framework for engineered mechanotransduction.

previous DNA MPs have made use of force-induced strand separation to reveal cryptic domains amenable for downstream reactivity, such as in situ dimerization or for ex situ amplification[20,21]. Fully protein-based sensors that mimic Notch signaling offer additional strategies to encode both receptor specificity and force responsiveness, yet these approaches typically require labor-intensive protein engineering, including insertion-site optimization to maintain receptor localization and biological function—thereby limiting accessibility, modularity, and scalability[22,23].

We hypothesized that aptamers—short single-stranded DNA sequences (ssDNA) selected for high-affinity binding to specific cell surface receptors—could serve as a versatile and new family of synthetic mechanosensors. Unlike classical mechanotransduction targets such as integrins or cadherins, many aptamer-accessible receptors are not canonically associated with force transmission. However, given their roles in signaling, trafficking, and cell migration, and since cells continuously recycle surface proteins for quality control, these receptors may transiently interact with the cytoskeleton or become indirectly coupled to intracellular force-generating machinery—potentially enabling mechanical engagement through previously unrecognized mechanisms. Indeed, a recent proof-of-concept using the CI-M6PR-targeting aptamer demonstrated that an aptamer can function as a force-reception module to sense weak endocytic forces (~ 4 pN)[24] and then translate this force cue into intracellular signals via DNA-mediated receptor dimerization[20]. However, that system required soft DNA hairpins suffering from dynamic breathing behavior, which limits force resolution and complicates integration with DNA reaction circuits. More generally, the design space for aptamer-based force sensors remains largely unclear—particularly in terms of cell-type specificity, force thresholding, and mechanistic understanding of how force is generated at the aptamer-receptor interface. Importantly, because aptamers are nucleic acids, they are inherently compatible with DNA-based strand displacement reactions and programmable logic circuits[25–27]. This opens opportunities for dynamic and autonomous regulation of mechanosensing—features that are difficult to achieve with conventional peptide- or protein-based systems like RGD-integrin probes or SynNotch receptors.

Here, we present a modular and fully programmable mechanosensing platform built entirely from nucleic acids, in which DNA aptamers function as synthetic receptor ligands embedded within force-responsive DNA MPs. This approach combines the receptor specificity of aptamers with the tunable force thresholds of DNA mechanics, enabling the systematic exploration of non-canonical and previously unrecognized force-generating surface receptors across diverse cell types. Using this platform, we elucidate the magnitude of noncanonically transmitted forces through DNA MP geometry, and robust cell-type selectivity dictated by aptamer-receptor binding. In contrast to conventional integrin-targeting RGD probes, our aptamer MPs display distinct force transmission behaviors, revealing mechanistic insights into how specific receptor contexts modulate mechanical engagement. Going further, leveraging the nucleic acid nature of aptamers, we integrate these probes with upstream DNA-based reaction networks to achieve temporal gating and switchable mechanosensitivity. This platform establishes a general framework for constructing adaptive, programmable mechano-interfaces and expands the synthetic biology toolkit for force-guided cell control.

## Results

### General concept for aptamer-based mechanosensing platform
Our aptamer-based MPs are composed of three modular elements: a terminal aptamer that selectively binds a target receptor, a DNA duplex that acts as a tunable force-sensitive element, and a fluorophore–quencher pair that converts mechanical unfolding into a fluorescence signal (Fig. 1). The aptamer functions as a programmable ligand, enabling receptor-specific engagement and ensuring that only receptor-expressing cells generate detectable force signals. To demonstrate the versatility of this platform, we employ five distinct and frequently used aptamers targeting diverse cell-surface receptors across five different cell lines. Moreover, the nucleic acid-based architecture of the aptamer permits dynamic control of receptor recognition via complementary blocker strands, which can be regulated through DNA-based reaction networks to achieve multi-layered regulatory complexity, such as temporally gated mechanosensing. For surface functionalization, we immobilized biotinylated MPs onto streptavidin-coated glass slides and co-immobilized biotinylated RGD peptides to support baseline integrin-mediated cell adhesion. Supported lipid bilayer-based surface density calibration assays reveal that the densities of different aptamer MPs are in the range of 3500–4500

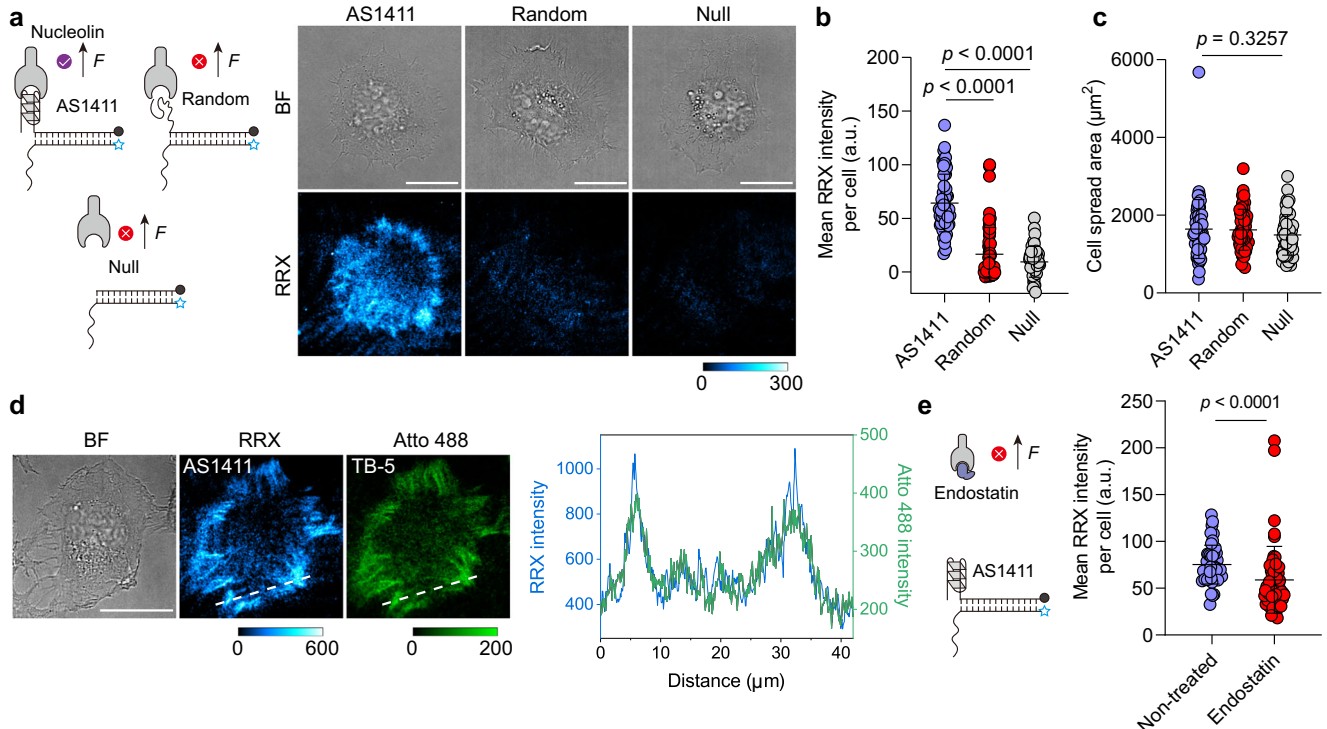

**Fig. 2 | Specific aptamer–receptor recognition is required for mechanosensing by aptamer-based MPs. a** Schematic and representative TIRF images of HeLa cells cultured on surfaces functionalized with AS1411 aptamer, a randomized sequence, or no ligand. **b, c** Quantification of mean fluorescence intensity per cell (**b**) and cell spreading area (**c**) for HeLa cells on each surface. $n = 54$ cells from 3 replicates. **d** Representative brightfield and fluorescence images (TIRF mode) of HeLa cells cultured on surfaces functionalized with a 1:1 ratio of AS1411 and TB-5 MPs. Line profile shows colocalization of AS1411 and TB-5 mechanosignals. **e** Scheme and quantification of mean fluorescence intensity per cell of HeLa cells without and with endostatin blocking (25 μg/mL) on AS1411 MP surfaces. $n = 45$ (control), 54 (blocked) cells from 3 replicates. Statistics (**b, c**): Kruskal-Wallis test with Dunn's multiple comparisons, **e** Two-tailed, Mann-Whitney test. All graphs, except (**d**) are presented as mean ± s.d. a.u., arbitrary units. Scale bars = 20 μm. Source data are provided as a Source Data file.

molecules/μm², and the overall stoichiometric ratio of MPs to RGD is ~3:1 (Supplementary Fig. 1, Supplementary Table 2).

## Force transmission of aptamer MPs requires specific aptamer-receptor recognition

To investigate the role of aptamer–receptor recognition in mechanosensing, we designed a MP incorporating the AS1411 aptamer, which binds nucleolin (dissociation constant $K_d = 69.1$ nM)[28]—a membrane-associated receptor overexpressed in many cancer cells[29]. We selected nucleolin-rich HeLa cells as the test system and used an AS1411 MP with a zipper-type duplex designed to rupture under a low force tolerance ($T_{tol}$) of 12 pN. Subsequently, we seeded cells onto the MP-functionalized surface and imaged them by total internal reflection fluorescence (TIRF) microscopy after 4 h incubation at 37 °C in 1 % FBS/ DMEM medium. Two controls highlight the successful specific activation: One control bears a randomized sequence without affinity to nucleolin, and one lacks any ligand (Fig. 2a). As expected, only the AS1411 MP generates significant fluorescence, confirming that specific aptamer-receptor binding enables force transmission (Fig. 2a, b). Neither control produces signal, confirming the absence of any non-specific interactions, such as from dye-membrane interactions (Fig. 2b). Interestingly, cell spreading area and cell aspect ratio are similar for the three surfaces (AS1411 MP, random MP and negative control) within the 4 h observation period, indicating that the AS1411-nucleolin couple does not promote cell adhesion (Fig. 2c, Supplementary Fig. 2).

Next, we validate that the AS1411 MP specifically engages the nucleolin receptor by performing colocalization and competitive binding assays. To this end, we employed the TB-5 aptamer[30], which binds to a distinct epitope on nucleolin compared to AS1411, and co-immobilized TB-5 and AS1411 MP on the surface at a 1:1 ratio. The mechanosignals from both MPs colocalize, confirming that they both engage nucleolin (Fig. 2d). To further confirm specificity, we blocked the AS1411–nucleolin interaction using endostatin, a natural ligand for nucleolin[29], which reduces the AS1411-derived mechanosignals by 21.7% (Fig. 2e). Flow cytometry corroborates this finding, showing a 23.2% decrease in AS1411 binding to HeLa cells in the presence of endostatin (Supplementary Fig. 3). Together, these results demonstrate that force transmission through the AS1411 MP originates from specific aptamer–receptor recognition.

## Aptamer MPs reveal force magnitudes across noncanonical mechanoreceptors

To compare the force magnitude of aptamer-targeted receptors with common RGD-targeted integrin[21], we designed each MP with two distinct force geometries. In the unzipping mode, the duplex opens base-by-base, resulting in a low $T_{tol}$ of 12 pN. In contrast, the shearing mode applies force across multiple base pairs near the duplex termini, yielding a higher $T_{tol}$ of 54 pN based on sequence design (Fig. 3a)[15].

Comparing first both RGD MPs, the RGD-unzipping MP produces markedly stronger fluorescence signals than the RGD-shearing MP, particularly at peripheral focal adhesions (FAs) and at central regions indicative of early-stage adhesion events (Fig. 3b). This pattern reflects the lower $T_{tol}$ of the unzipping mode, which enables detection of nascent, low-force adhesion events that fail to activate RGD-shearing MP. Although altering the force geometry modulates the average mechanoresponse pattern per cell (Fig. 3c), the overall response based on the integrated intensity remains indistinguishable and independent of the MP geometry (Fig. 3d). The integrated intensity reflects the cumulative mechanosignal across the entire cell footprint and is thus

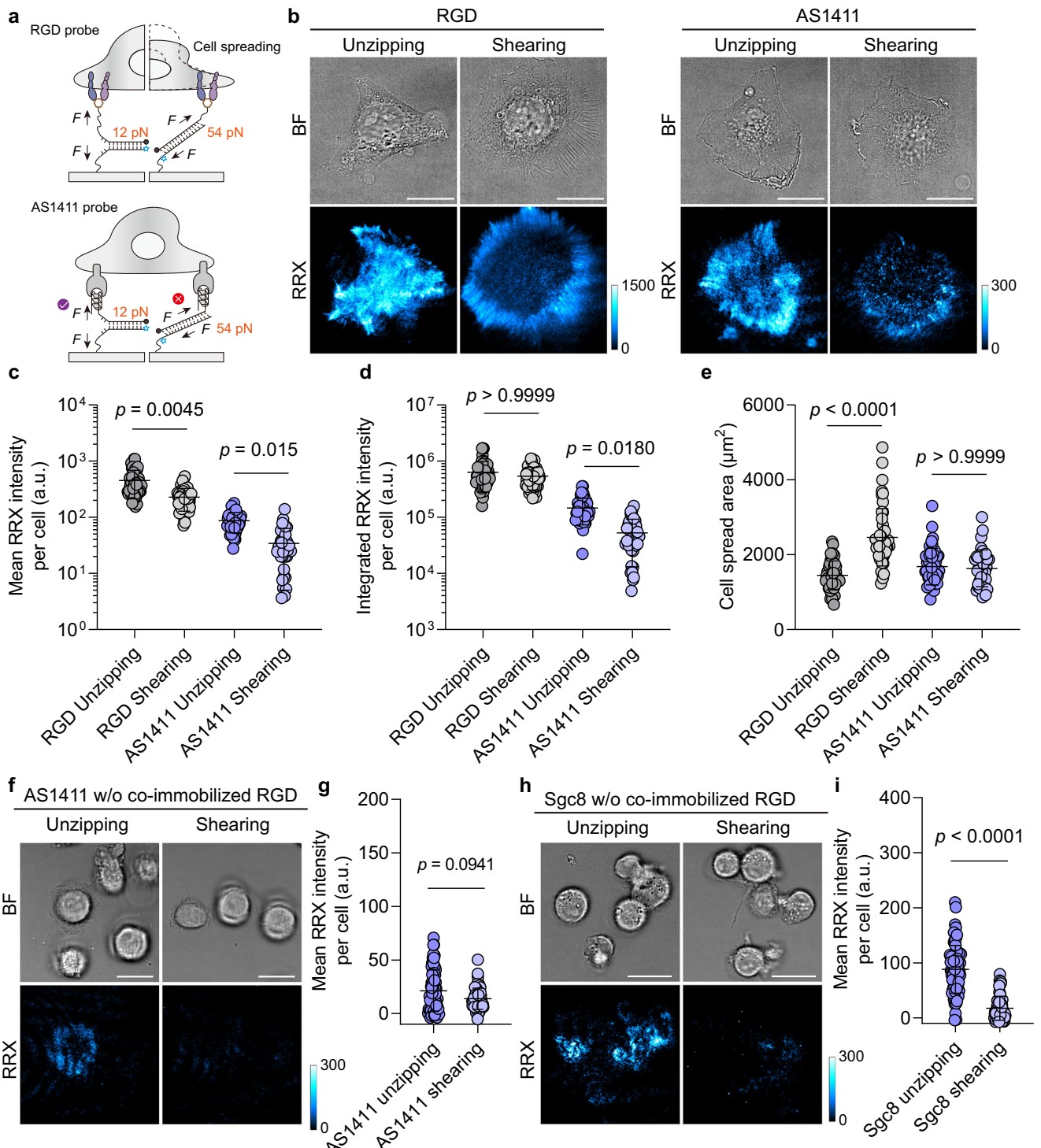

not only regulated by intrinsic $T_{tol}$ of the probe but inherently dependent on cell area[19]. This apparent convergence arises because RGD-integrin linkages are capable of transmitting single-molecule forces that exceed both $T_{tol}$ values (>54 pN), and in some cases can even rupture the streptavidin-biotin linkage (>100 pN)[31]. However, the generation of such high forces is contingent on adequate mechanical resistance from the substrate—typically requiring 33-44 pN counter-force to sustain cytoskeletal maturation and stable adhesion[32]. As a result, shearing-mode MPs promote cell spreading and increase overall force generation, thereby blurring the difference in integrated mechanoresponse between both geometries (Fig. 3e)[19].

In contrast, AS1411 MPs show a pronounced distinction between unzipping and shearing geometry. The unzipping mode consistently produces stronger signals in both mean and integrated intensity (Fig. 3b–d). Two factors account for this behavior: First, the AS1411-nucleolin couple transmits substantially lower force (consistently below 54 pN) than RGD-integrin engagement. Second, unlike integrins, the AS1411-nucleolin pair alone does not promote cell spreading (Figs. 2c, 3e). This property makes the shearing mode of aptamer MPs particularly well-suited for tandem integration with other DNA nanodevices[33,34], where mechanical stability is essential to avoid unintended force-induced detachment.

**Fig. 3 | Aptamer MPs reveal force magnitudes across noncanonical mechanoreceptors. a** Scheme of the mechanosensing through unzipping and shearing geometries in RGD and AS1411 MPs. **b** Representative brightfield and fluorescence images of HeLa cells cultured on surfaces functionalized with RGD or AS1411 MPs featuring different force geometries. Quantification of (**c**) mean fluorescence intensity per cell, (**d**) integrated fluorescence intensity per cell, and (**e**) cell spreading area for HeLa cells on each MP type and geometry. $n = 49, 47, 45, 36$ (left to right) cells from 3 replicates. Statistics: Kruskal-Wallis test with Dunn's multiple comparisons. Note that both mean and integrated intensity are shown because mean intensity mainly captures the $T_{tol}$-dependent shifts, while integrated intensity is cross-regulated by both the intrinsic $T_{tol}$ of the probe and the cell spreading area. **f** Representative brightfield and fluorescence images of HeLa cells cultured on surfaces functionalized with AS1411 MPs alone, featuring different force geometries. No co-immobilized RGD-biotin is present to support baseline cell adhesion. Most HeLa cells remain non-adherent and fail to activate the unzipping AS1411 MPs. Only few cells adhering to the surface via nonspecific interactions can activate MPs. **g** Quantification of mean fluorescence intensity per cell for HeLa cells on each geometry. $n = 68, 67$ cells from 3 replicates. Statistics: Two-tailed, Mann-Whitney test. **h** Representative brightfield and fluorescence images of HepG2 cells cultured on surfaces functionalized with Sgc8 MPs alone featuring different force geometries. No co-immobilized RGD-biotin is present to support baseline cell adhesion. Non-spread HepG2 cells activate Sgc8 MPs. **i** Quantification of mean fluorescence intensity per cell for HepG2 cells on each geometry. $n = 71, 68$ cells from 3 replicates. Statistics: Two-tailed, Mann-Whitney test. All graphs are presented as mean ± s.d. a.u., arbitrary units. Scale bars = 20 μm. Source data are provided as a Source Data file.

Since the above results suggest that aptamers cannot promote cell adhesion, it raises the question to what extent baseline adhesion is required for aptamer MPs to function. To address this, we prepared surfaces functionalized solely with aptamer MPs without co-immobilized RGD to investigate mechanosensing under minimal adhesion conditions. HeLa cells are largely unable to spread on surfaces presenting AS1411 MPs alone, regardless of whether unzipping or shearing MPs are used (Fig. 3f). Compared with RGD co-immobilized conditions, AS1411 unzipping-mediated mechanosignals are significantly reduced to below 24.4 % (Fig. 3g), implying that baseline cell adhesion is a prerequisite for AS1411-based force sensing.

Considering that we aim to develop receptor-orthogonal aptamer MPs, we further investigated the Sgc8-aptamer as a potential MP for the protein tyrosine kinase 7 (PTK7), using HepG2 cells that overexpress this receptor. Interestingly, Sgc8 MPs-functionalized surfaces exhibit a somewhat different behavior (Fig. 3h). Even though Sgc8 MPs alone also fail to promote HepG2 cell spreading, robust mechanoactivation occurs (Fig. 3i). Even in a non-spread, quasi-suspended state (see brightfield images), HepG2 cells effectively activate Sgc8 unzipping MPs, yielding ring-like mechanosignals. This indicates that for Sgc8-PTK7, baseline cell adhesion is not a prerequisite for mechanoactivation. The differential adhesion dependence of these two aptamer MPs suggests that they transmit forces through fundamentally distinct mechanisms, which we will further discuss below.

## Source of forces associated with noncanonical mechanoreceptors

Although nucleolin is a well-established molecular target[35,36], it has not been previously associated with the exertion of mechanical force on the extracellular environment. Our data reveal distinct differences to integrin-targeting MPs, prompting us to investigate the source of nucleolin-associated forces. To test whether aptamer-receptor internalization contributes to force generation, we treated HeLa cells with pathway-specific inhibitors. Neither chlorpromazine (CPZ, 28 μM), which inhibits clathrin-mediated internalization[37], nor methyl-β-cyclodextrin (MβCD, 1.9 mM), which inhibits caveolin- and lipid-raft-mediated uptake[38], alters the mechanosignal intensity (Fig. 4a, b). In contrast, treatment with 5-(N-ethyl-N-isopropyl) amiloride (EIPA, 100 μM), an inhibitor of macropinocytosis[39], reduces mechanosignals by 29.8 % (Fig. 4b). This implicates macropinocytosis in the mechanoactivation of the AS1411 MP, consistent with previous findings that cancer cells take up free AS1411 aptamers predominantly via macropinocytosis[40]. Flow cytometry further supports this conclusion, showing a 76.6 % reduction in the uptake of free AS1411 aptamers by HeLa cells in the presence of EIPA (Supplementary Fig. 4). Notably, compared to the free state in solution, the inhibitory effect of EIPA decreases by 46 % when AS1411 is immobilized as an MP on a surface. These results suggest that while macropinocytosis contributes to force generation, it may not be the main source of nucleolin-mediated force.

Interestingly, previous studies reported molecular associations between nucleolin and multiple focal adhesion-related or cytoskeletal components, including integrins $\alpha_v\beta_3$[41], $\alpha_5\beta_1$[42], and $\beta_1$[43], focal adhesion protein talin[44], and myosin heavy chain 9[45]. To determine whether nucleolin-mediated forces spatially correlate with FA structures, we transfected HeLa cells with a plasmid for constitutive expression of monomeric enhanced green fluorescent protein (mEGFP)-paxillin and visualized the actin cytoskeleton with phalloidin staining. Notably, nucleolin-generated forces are clearly localized at peripheral FA sites marked by paxillin, suggesting a spatial and functional association between nucleolin and FAs (Fig. 4c, d, Supplementary Fig. 5). Based on this, we hypothesized nucleolin may indirectly couple to the actomyosin cytoskeleton to transmit mechanical force (Fig. 4e). To identify the cytoskeletal components responsible for this transmission, we treated cells with specific inhibitors after allowing 30 min of initial adhesion[16,46]. Inhibiting actin polymerization with cytochalasin D (10 μM) moderately reduces mechanosignals by 38.9 % (Fig. 4f, Supplementary Fig. 6). In contrast, blocking myosin II ATPase activity with blebbistatin (25 μM) causes a pronounced 77.5 % decrease. Together, these results demonstrate that nucleolin-mediated forces primarily depend on myosin-driven contractility, which may be relayed through FAs or other physical linkages independently coupled to the actomyosin network before reaching nucleolin.

To understand the differential dependence of Sgc8 aptamer MPs on baseline adhesion (see above, Fig. 3f, h), we investigated the underlying source of Sgc8-PTK7 forces. Unlike the FA-like patterns observed for nucleolin, PTK7-activated mechanosignals display spatially distinct, ring-like force patterns with submicron size (Fig. 4g). Those are consistent with a macropinocytic origin, as clathrin- and caveolin-mediated internalization typically generates vesicles in the nanometer range[47]. To test this hypothesis, we inhibited macropinocytosis using EIPA (100 μM), which substantially reduces mechanoactivation by 51.9 %, whereas CPZ (28 μM) and MβCD (1.9 mM) inhibiting clathrin- and caveolin-mediated endocytosis, have no effect (Fig. 4g, h, Supplementary Fig. 7). To further examine the spatial organization of these signals, we stained the plasma membrane and performed TIRF microscopy. As expected, the ring-like mechanofluorescence predominantly localizes to the peripheries of dark membrane invaginations, consistent with force generation during membrane ruffling in early-stage macropinocytosis (Fig. 4i, j, Supplementary Fig. 8)[48].

To understand the cytoskeletal origin of this force generation, we next examined the roles of actin polymerization and myosin action during different macropinocytic phases. Actin polymerization is critical for driving membrane ruffling and protrusion formation during the early-stage macropinocytosis, whereas myosins contribute to actin network contraction at the base of ruffles as the membrane closes to form macropinosomes[49]. Treatment with cytochalasin D (10 μM, actin polymerization inhibitor) and blebbistatin (25 μM, myosin II inhibitor) reduces mechanosignals by 56.1 % and 32.6 %, respectively, suggesting that force generation is dominated by actin polymerization rather than myosin-driven contraction (Fig. 4k, Supplementary Fig. 9). Phosphoinositide 3-kinase (PI3K)

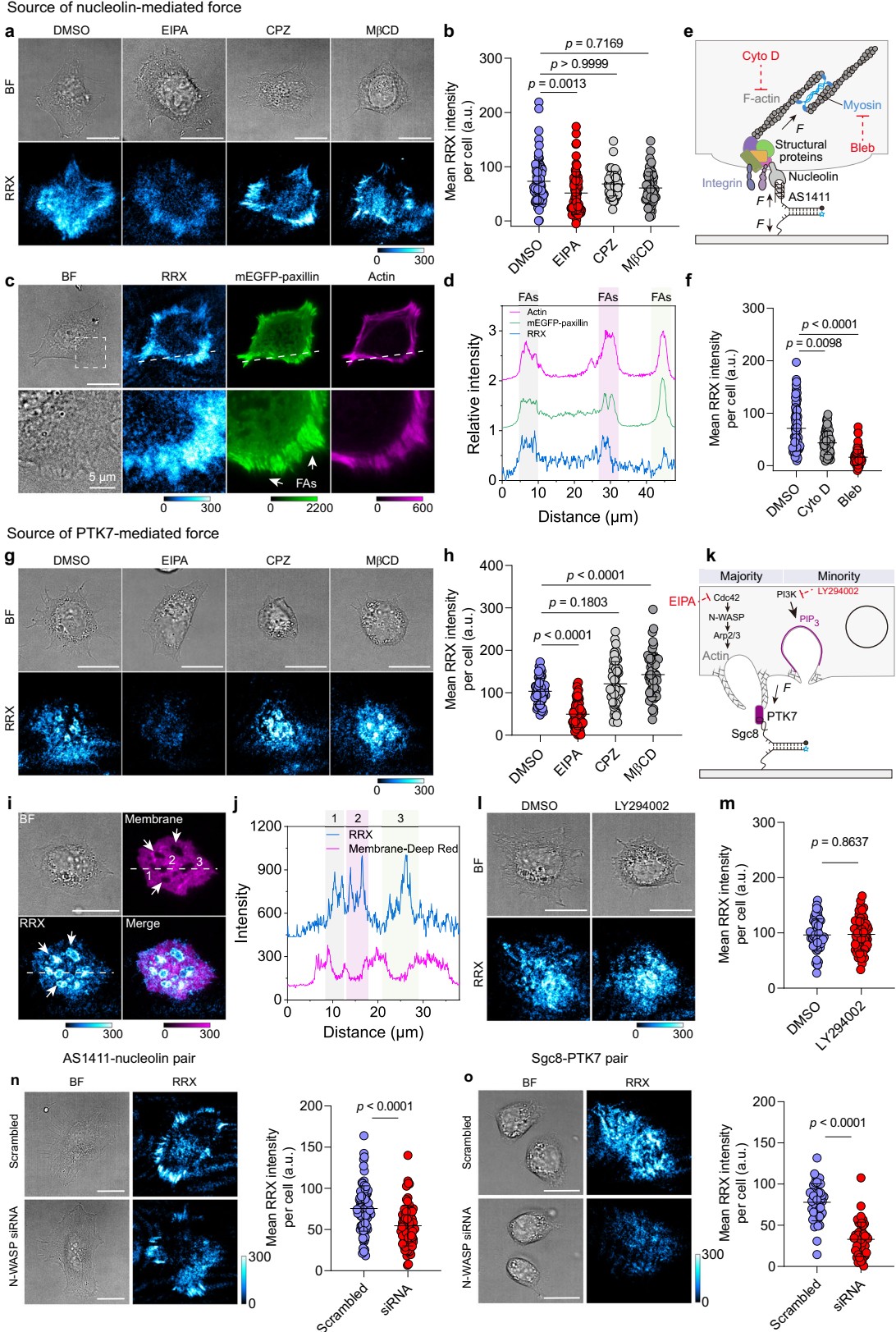

Source of nucleolin-mediated force

Source of PTK7-mediated force

AS1411-nucleolin pair

Sgc8-PTK7 pair

catalyzes membrane sealing by generating PIP$_3$ at the ruffle bases, thereby recruiting effectors for actin remodeling and vesicle closure[50]. While PI3K activity is essential for membrane sealing, it is dispensable for ruffle formation[51]. Consistent with this, inhibition of PI3K using LY294002 (50 μM) does not affect mechanosignals, further confirming that PTK7-mediated forces originate primarily from

early-stage membrane ruffling during macropinocytosis with minimal contribution from the sealing phase (Fig. 4l, m).

To further distinguish between two force transmission mechanisms, we knocked down the orthogonal regulator neuronal Wiskott-Aldrich syndrome protein (N-WASP). As a critical cytoskeletal regulator, N-WASP interacts with the Arp2/3 complex to drive branched

**Fig. 4 | Aptamer MPs reveal force origins associated with noncanonical mechanoreceptors. a** Representative brightfield and fluorescence images of HeLa cells treated with internalization inhibitors on AS1411 MP surfaces. **b** Corresponding quantification of mean fluorescence intensity per cell for HeLa cells. $n = 70, 70, 54, 54$ (left to right) cells from 3 replicates. Kruskal-Wallis test with Dunn's multiple comparisons. **c** Representative images of HeLa cells on AS1411 MP surfaces showing brightfield, mechanosignal, mEGFP-paxillin, and phalloidin-stained actin. **d** Line profile from (**c**) shows intensity profiles of mechanosignals, mEGFP-paxillin, and actin. **e** Schematic showing the source of nucleolin-mediated forces. **f** Quantification of mean fluorescence intensity per cell following cytoskeletal inhibition. $n = 72$ cells from 3 replicates. Kruskal-Wallis test with Dunn's multiple comparisons. **g** Representative brightfield and fluorescent images of HepG2 cells treated with internalization inhibitors on Sgc8 MP surfaces. **h** Corresponding quantification of mean fluorescence intensity per cell. $n = 54$ cells from 3 replicates. One-way ANOVA with Bonferroni post-hoc tests. **i** Representative images showing plasma membrane staining, mechanosignals, and brightfield of HepG2 cells incubated on Sgc8 MP surfaces. **j** Line profile from (**i**) reveals spatial colocalization of mechanosignals at the inner edge of membrane invaginations. **k** Schematic showing the source of PTK7-mediated forces. **l** Brightfield and fluorescence images of HepG2 cells treated with PI3K inhibitor LY294002 on Sgc8 MP surfaces. **m** Corresponding quantification of mean fluorescence intensity per cell for HepG2 cells. $n = 54$ cells from 3 replicates. Unpaired, two-tailed Student's t-test. **n** Representative brightfield and fluorescence images of HeLa cells after N-WASP siRNA knockdown on AS1411 MP surfaces, and mean fluorescence intensity per cell. $n = 85, 83$ cells from 3 replicates for scrambled and siRNA groups. Unpaired two-tailed Student's t-test. **o** Representative brightfield and fluorescence images of HepG2 cells after N-WASP siRNA knockdown on Sgc8 MP surfaces, and mean fluorescence intensity per cell. $n = 45$ cells from 3 replicates. Unpaired two-tailed Student's t-test. All graphs, except (**d, j**), are presented as mean ± s.d. a.u., arbitrary units. Scale bar = 20 μm. Source data are provided as a Source Data file.

actin polymerization[52], thereby regulating essential cellular processes such as membrane ruffling, endocytosis, and cytoskeletal remodeling. Inhibition of the Arp2/3 complex has been shown to inhibit Arp2/3-mediated actin polymerization in phagocytic adhesion rings of macrophages but has little effect on integrin-mediated tensions in FAs[53]. The latter depends on myosin II-loaded stress fibers which lack the Arp2/3 complex. For the AS1411-nucleolin pair, compared with cells transfected with scrambled siRNA, N-WASP knockdown HeLa cells still exhibit a stripe-like force pattern, although the signal amplitude is reduced by 27.9% (Fig. 4n, Supplementary Fig. 10a). This result indicates that force transmission through the AS1411-nucleolin pair is weakly dependent on N-WASP. In contrast, for the Sgc8-PTK7 pair, N-WASP knockdown in HepG2 cells leads to an almost complete loss of the ring-like pattern, with a 57.8% reduction in signal intensity relative to scrambled controls, suggesting a strong dependence on N-WASP-mediated actin dynamics (Fig. 4o, Supplementary Fig. 10b). Moreover, the different effects of N-WASP knockdown in both cell lines closely mirror the results observed upon EIPA treatment (Fig. 4b, h). This is consistent with the fact that EIPA inhibits Rac1/Cdc42 activity, and Cdc42 is a key upstream activator of N-WASP[54]. Together, these results demonstrate that the AS1411-nucleolin and Sgc8-PTK7 pairs exhibit distinct dependencies on N-WASP and downstream Arp2/3-mediated branched actin polymerization, reflecting fundamentally different force transmission pathways.

## Versatility of aptamer MPs to customize cell-type-specific mechanoresponses

The specificity of aptamer-receptor recognition enables the modular customization of MPs for cell-type-specific mechanosensing. We demonstrate this principle using three aptamer MPs—Sgc8, MUC1 S2.2, and SYL3c—which target PTK7 ($K_d = 0.85$ nM)[55], mucin-1 ($K_d = 0.12$ nM)[56], and epithelial cell adhesion molecule (EpCAM, $K_d = 38$ nM)[57], respectively. We tested each MP on both receptor-high-expressing and receptor-low-expressing cell lines, selected based on differential receptor expression profiles confirmed by flow cytometry (Supplementary Fig. 11). Mucin-1-high-expressing HeLa cells strongly activate the MUC1 S2.2 MP, whereas HepG2 cells with substantially lower levels of mucin-1 show minimal response (Fig. 5a,b). Conversely, HeLa cells fail to activate the SYL3c MP due to low EpCAM receptor levels. This contrasts with EpCAM-high-expressing MDA-MB-231 cells (Fig. 5c, d). Similarly, HepG2 cells generate stronger activation of the Sgc8 MP than A549 cells, consistent with their higher PTK7 expression (Fig. 5e, f). Interestingly, beyond expression level-dependent signal magnitude, different cell types expressing the same receptor can exhibit distinct, cell-type-dependent force patterns. HeLa cells express PTK7 at an intermediate level (Supplementary Fig. 11e, f). Mechanosensing using Sgc8 MPs reveals a stripe-like force pattern in contrast to the characteristic ring-like mechanosignals observed in HepG2 cells (Fig. 5e, f). These results indicate that different cell types may transmit forces through the same receptor via distinct pathways, a phenomenon that cannot be inferred from receptor expression levels alone. Collectively, the high differential selectivity of aptamer-receptor MP pairs highlights the broad adaptability of our platform for sensing mechanoresponses across diverse cell types and receptor classes.

Motivated by the need for selective and precise mechanoresponse in complex multicellular environments, we extend our validation beyond simple single-cell-type systems. To this end, we labeled receptor-high-expressing and receptor-low-expressing cells with CellTrace Deep Red and Violet dyes, respectively, and co-seeded them in wells containing the corresponding aptamer MPs to assess cell-type-specific mechanoresponse (Fig. 5g). Indeed, mechanoactivation of the Sgc8 MP occurs exclusively in the Deep Red-labeled PTK7-high-expressing HepG2 cells, while the Violet-labeled PTK7-low-expressing A549 cells exhibit much lower signals (Fig. 5h, i). Switching to the MUC1 S2.2 MP leads to strong mechanosignals in the Deep Red-labeled mucin-1-high-expressing HeLa cells, whereas Violet-labeled mucin-1-low-expressing HepG2 cells show minimal signals (Fig. 5j, k). These results underscore the utility of aptamer MPs as customizable mechanical elements for precise, cell-type-specific mechanochemical transduction in heterogeneous cell populations.

## Aptamer MPs enable programmable mechanoresponses via integration with DNA-based reaction networks (DRNs)

Natural mechanotransduction involves temporally orchestrated biochemical signals that guide adaptive cellular responses. Aptamers, unlike protein or peptide ligands, offer the unique advantage that their DNA structure can be seamlessly integrated with DRNs to emulate this dynamic regulation. Importantly, aptamers can be manipulated directly at their binding site with DRNs without detaching them from the underlying surface by for instance breaking the force-sensitive duplex, thus ensuring their correct configuration and integrity.

To introduce temporal control over mechanoactivation, we regulate aptamer accessibility using designer blocker and activator strands that can be designed with tunable affinities based on base-pairing principles (Fig. 6a). We first use a blocker fully complementary to AS1411 to completely prevent binding to nucleolin. This blocker also contains a 5-nucleotide (nt) toehold, enabling its subsequent removal via a toehold-mediated strand displacement reaction (SDR) triggered by an activator strand. In this design, mechanoactivation becomes conditional on the possibility for aptamer-receptor engagement, which can be dynamically regulated through SDR. Cell-free surface assays confirm robust and reversible aptamer blocking (Supplementary Fig. 12). The blocker quenches >90% of a fluorescently labeled, surface-tethered AS1411 aptamer within 30 min, indicating effective blocking. Subsequent medium exchange with the equimolar DNA activator restores ~50% of the signal within 30 min with a reaction rate of $2.14 \times 10^{-2}$ min$^{-1}$, confirming aptamer reactivation (see Methods, Supplementary Fig. 12a–d, Supplementary Table 3). We next validate

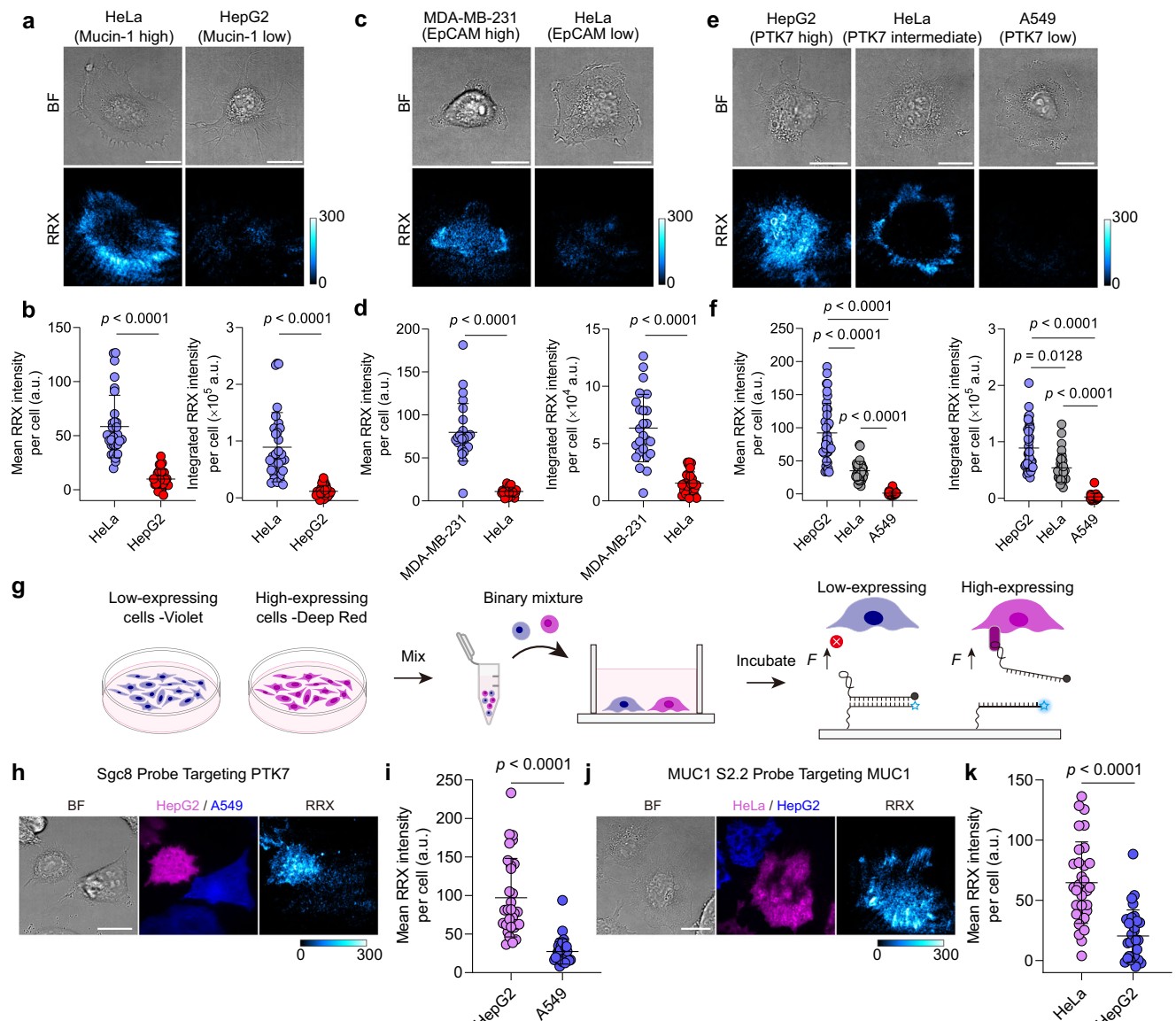

**Fig. 5 | Customized cell-type-specific mechanoactivation enabled by aptamer-receptor recognition. a** Representative images of HeLa (mucin-1-high-expressing) and HepG2 (mucin-1-low-expressing) cells on MUC1 S2.2 MP surfaces. **b** Corresponding quantification of mean and integrated fluorescence intensity per cell of HeLa and HepG2 cells on MUC1 S2.2 MP surfaces. $n = 31, 27$ cells from 3 replicates. Statistics: Two-tailed, Mann-Whitney test. Note that both mean and integrated intensity are shown here because different cell lines may have distinct basal morphologies with different cell area. **c** Representative images of MDA-MB-231 (EpCAM-high-expressing) and HeLa (EpCAM-low-expressing) cells on SYL3c MP surfaces. **d** Corresponding quantification of mean and integrated fluorescence intensity per cell of MDA-MB-231 and HeLa cells on SYL3c MP surfaces. $n = 25, 27$ cells from 3 replicates. Statistics: Two-tailed, Mann-Whitney test. **e** Representative brightfield and fluorescence images of HepG2 (PTK7-high-expressing), HeLa (PTK7-intermediate-expressing) and A549 (PTK7-low-expressing) cells on Sgc8 MP surfaces. **f** Corresponding quantification of mean and integrated fluorescence

intensity per cell of HepG2, HeLa and A549 cells on Sgc8 MP surfaces. $n = 36, 31, 27$ cells from 3 replicates. Statistics: Kruskal-Wallis test with Dunn's multiple comparisons. **g** Schematic of co-culture assay design. Receptor-high-expressing and -low-expressing cells were pre-labeled with CellTrace Deep Red and Violet dyes, respectively, mixed 1:1, and co-seeded onto aptamer MP-functionalized surfaces. **h** Representative images of HepG2 (PTK7-high-expressing, Deep Red) and A549 (PTK7-low-expressing, Violet) cells co-cultured on Sgc8 MP surfaces. **i** Quantification of mechanosignals in HepG2 vs A549 cells. $n = 30$ cells from 3 replicates. Statistics: Two-tailed, Mann-Whitney test. **j** Representative images of co-cultured HeLa (mucin-1-high-expressing, Deep Red) and HepG2 (mucin-1-low-expressing, Violet) cells on MUC1 S2.2 MP surfaces. **k** Corresponding quantification of mechanosignals as mean fluorescence intensity per cell. $n = 31$ cells from 3 replicates. Statistics: Unpaired, two-tailed student's t-test. All graphs are presented as mean ± s.d. a.u., arbitrary units. Scale bars = 20 μm. Source data are provided as a Source Data file.

this mechanism under live-cell conditions for two consecutive suppression-recovery cycles. In the first cycle, adding 200 nM blocker 30 min after seeding suppresses AS1411-nucleolin mechanoactivation by 75.1 %. Subsequent medium exchange with 200 nM activator restores 82.5 % of the original signal, consistent with aptamer re-engagement and functional recovery of mechanosensing (Fig. 6b). The second blocker-activator cycle closely mirrors the suppression-recovery behavior observed in the initial cycle (Supplementary

Fig. 12e). The overall higher signal level can be attributed to cumulative mechanoactivation accumulated during the additional activation period.

Going a step beyond external switching, we next introduce more advanced enzyme-catalyzed DRNs upstream of mechanoactivation to autonomously program the temporal accessibility of aptamer MPs[26,27]. To this end, we replace the DNA blocker with an RNA analogue. This RNA blocker can be selectively degraded by RNase H only when

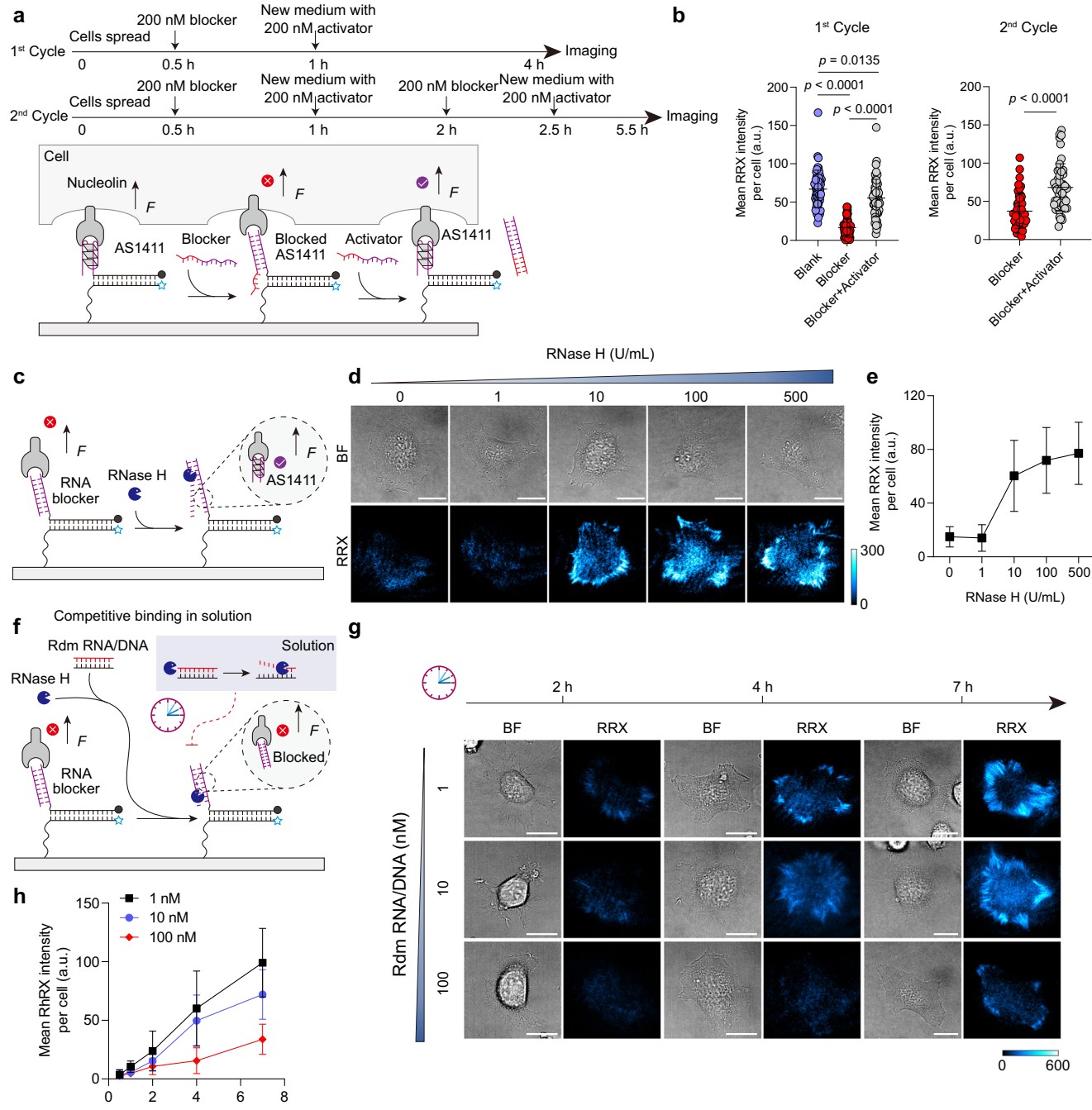

**Fig. 6 | Programmable mechanoactivation enabled by upstream DRNs.**
**a** Schematic of switchable mechanosensing using SDRs for two suppression-recovery cycles. In the 1st cycle, HeLa cells were seeded on surfaces grafted with AS1411 MP for 30 min, followed by addition of 200 nM DNA blocker to inhibit aptamer-receptor interaction. After another 30 min, 200 nM DNA activator was introduced to displace the blocker, restoring mechanosensing. Mechanosignals were quantified after 3 h. In the 2nd cycle, another round of blocker was introduced 1 h after adding activator in the 1st cycle. After another 30 min, a second-round activator was introduced. The system was incubated for 3 h prior to imaging. **b** Quantification of mean fluorescence intensity per cell of HeLa cells on AS1411 MP surfaces after treatment for two suppression-recovery cycles. 1st cycle, $n = 54$ cells from 3 replicates. One-way ANOVA with Bonferroni post-hoc tests. 2nd cycle, $n = 48$ cells from 3 replicates. Two-tailed, Mann-Whitney test. **c** Schematic of enzyme-catalyzed mechanoactivation. Aptamer MPs are initially blocked by complementary RNA strands. Addition of RNase H at varying concentrations (0–500 U/mL)

degrades the RNA in the RNA–DNA duplexes, resetting the aptamer configuration and restoring mechanosensing. Cells were imaged after 4 h of incubation. **d** Representative brightfield and fluorescence images of HeLa cells on RNA-blocked AS1411 MP surfaces following addition of different RNase H concentrations. **e** Corresponding quantification of mean fluorescence intensity per cell. $n = 36$ cells from 3 replicates. **f** Schematic of delayed mechanoactivation. Aptamer MPs are RNA-blocked and exposed to 10 U/mL RNase H. Increasing concentrations of freely diffusing RNA–DNA decoy duplexes (1–100 nM) compete for RNase H activity, thereby delaying aptamer reactivation. **g** Representative brightfield and fluorescence images of HeLa cells on RNA-blocked AS1411 MP surfaces after adding 10 U/mL RNase H and varying RNA/DNA duplex concentrations. **h** Corresponding quantification of mean fluorescence intensity per cell. Reaction medium: 1 % FBS, 1 % P/S, 100 μg/mL actin protein, DMEM. $n = 27$ cells from 3 replicates. All graphs are presented as mean ± s.d. a.u., arbitrary units. Scale bars = 20 μm. Source data are provided as a Source Data file.

bonded as RNA-DNA duplex at the MP. This degradation releases the aptamer and restores mechanoactivation within programmable time windows (Fig. 6c). In this context, RNase H acts as a biochemical gatekeeper. Additional temporal control can be gained by introducing freely diffusing RNA-DNA duplexes as decoy substrates, which compete with the RNA-blocked MP for RNase H activity.

The kinetics of the timer circuit for MP activation depend on the DRN parameters. First, the RNase H concentration can be used to control the time for mechanoactivation to occur. Monitoring the aptamer reconfiguration kinetics with simple fluorescence assays reveals RNase H concentration-dependent reaction rates of $0.71 \times 10^{-2}\,min^{-1}$ at 10 U/mL and $5.26 \times 10^{-2}\,min^{-1}$ at 100 U/mL (see Methods, Supplementary Fig. 13a, Supplementary Table 3). When it comes to force sensing, low RNase H concentrations ($\leq 1$U/ml) fail to initiate measurable mechanoresponse within 4 h. At 10 U/mL, mechanoactivation is restored to 78.2%, while 100 U/mL yields near-complete recovery, consistent with efficient RNA blocker degradation (Fig. 6d, e). To further tune the system, we then fixed RNase H at 10 U/mL and introduced the RNA-DNA decoy (Fig. 6f). The introduction of the decoy strand exerts a clear, dose-dependent inhibitory effect on the reaction rate of RNA blocker degradation (Supplementary Fig. 13b, Supplementary Table 3). Consistent with this trend, 10 nM decoy is sufficient to slightly delay the mechanoresponse, whereas 100 nM nearly abolishes the response within 4 h and a significant response is observed only at 7 h (Fig. 6g, h, Supplementary Fig. 13c).

Because RNA-RNaseH modules can be extended into more complex signaling DRNs[25], our aptamer manipulation strategy opens doors for more complex temporal programs in the future. We note that similar programmable mechanoactivation could also be achieved using antibody- or nanobody-based probes that block receptor access via engineered protein blockers and activators with progressively tuned affinities. However, such protein-based systems would require extensive efforts, including site-directed mutagenesis, affinity optimization, and computational modeling to fine-tune the underlying protein–protein interactions. In contrast, aptamer systems offer a far more accessible and modular alternative, leveraging predictable base-pairing rules to design dynamic, multistage mechanical response circuits in a straightforward manner.

## Discussion

We have introduced aptamer-based MPs as a specific, modular, and programmable platform for mechanosensing that enables customizable mechanoresponses with cell-type selectivity. We find that aptamer–receptor recognition is a critical determinant of specific force transmission. Exploiting distinct receptor expression profiles on different cells, it is now possible to engineer cell-type-specific mechanoresponses that filter irrelevant mechanical stimuli and facilitate precise functional specialization. Such capabilities are essential for constructing complex behaviors in multicellular systems.

To contextualize the advantages of aptamer MPs, it is instructive to compare them with conventional ligand–receptor systems commonly used in mechanobiology. Although high-affinity ligands such as cRGDfK can achieve targeted, force-responsive delivery of therapeutic agents (e.g., anticancer drugs or antisense oligonucleotides) across $\alpha_v\beta_3$-positive cells[34,58], their promiscuity toward various integrin subtypes and the structural similarity among integrins limit specificity[18]. Moreover, RGD-integrin engagement promotes adhesion reinforcement, which blurs discrimination between different force geometries. In contrast, aptamers, as shown for the Sgc8-PTK7 interaction, can be functionally decoupled from adhesion processes, rendering the tension tolerance ($T_{tol}$) a dominant parameter in mechanoactivation. This enables the use of aptamer MPs as force-selective mechanical gates and allows their modular integration with other DNA-based nanodevices.

The excellent selectivity of aptamer MPs underscores their importance to interpret noncanonical mechanoreceptors outside of the classical mechanobiology realm. On the molecular level, we have elucidated two distinct force transmission behaviors: nucleolin-mediated forces arise from actomyosin contractility, whereas PTK7-mediated forces originate from membrane ruffling/actin polymerization during early-stage macropinocytosis. These two cases are not meant to be exhaustive but rather exemplify that noncanonical mechanoreceptors can engage in mechanical signaling via diverse, receptor-specific pathways. By proposing a systematic framework for receptor-specific mechanosensing, we lay the groundwork for future exploration of alternative mechanotransduction pathways. These uncovered mechanisms also expand the potential of aptamers as versatile force-reception modules for bidirectional mechano-biochemical signal transduction. Beyond relying on weak DNA hairpins to sense clathrin-mediated endocytic forces ($\sim 4$ pN)[20], cells can exert forces >12 pN on aptamer MPs. Such forces allow the irreversible exposure of cryptic ssDNA domains that can trigger downstream reactions such as transcription, hairpin chain reactions[21], or rolling circle amplification[59]. This expanded force range substantially broadens the design space for intelligent, force-responsive material interfaces capable of cellular adaptation.

The identification of noncanonical forces highlights the importance of revisiting the biological functions of both these aptamers and receptors from a mechanobiological perspective. Determining whether these receptors are directly coupled to classical adhesion structures or are independently linked to the same actomyosin network via intermediary components will be essential for uncovering more general principles. Whether similar interactions occur under native conditions remains to be elucidated. A prime candidate is the precursor of the nucleolin ligand endostatin, located in the NC1 domain of type XVIII collagen[60], which may serve as a native anchor for nucleolin despite potential differences in binding affinity. Addressing these questions could reveal how cryptic motifs within the ECM regulate cellular homeostasis through non-canonical mechanotransduction pathways. Meanwhile, caution is required when extrapolating from aptamer-receptor pairs to native ligand-receptor interaction, particularly regarding the immobilization state of the ligand, binding affinity, and bond mechanics, and the multivalency and cooperative effects inherent to native ligands. For example, the AS1411 aptamer binds nucleolin with a $K_d$ of 69.1 nM[28], which is lower than that of its native ligand endostatin (23.2 nM)[61], suggesting that our measurements may slightly underestimate the native capacity for tension generation. Finally, the generalizability of the aptamer-based platform provides opportunities to probe other established aptamer-receptor pairs with well-defined biological functions, such as PD-L1-targeting aptamers involved in immune checkpoint regulation[62,63], or CD28 costimulatory aptamers in T cells[64], to determine whether their biological activities are similarly regulated by mechanical force.

The all-nucleic-acid nature of aptamer MPs offers unique advantages by enabling seamless integration with the broader toolbox of DNA nanotechnology. Without detaching MPs (or ligands), we introduced two modes of precise upstream control over receptor engagement: Reversible switching via SDR and programmable timing via RNase H–driven DRNs. Unlike protein- or nanobody-based MPs, this programmability requires no protein engineering, expression, or purification, but only base-pairing logic. This molecular tunability allows mechanotransduction to transcend simple linear outputs and adopt complex, nonlinear, and time-resolved behaviors. Such temporal precision is especially vital in biological systems, such as morphogenesis and embryonic development, where cells interpret mechanical and biochemical signals in tightly orchestrated spatiotemporal patterns[7]. Moving forward, pioneering work using protein-ligand functionalized MPs has shown that strand displacement can control intracellular mechanics in 3D spheroids[65], although at the cost

of ligand release. Introducing aptamer-based MPs further enables non-invasive, tunable modulation without compromising probe integrity. However, long-term application in these environments still faces stability challenges. While modifications like peptide nucleic acids[66], phosphorothioate linkages[67], or our recent L-DNA strategy protect the MP structure[68], single-stranded aptamers remain inherently degradation-prone. Balancing their structural integrity with binding affinity is critical. In this context, using nuclease inhibitors such as actin or decoy DNA may be more effective[69,70]. Additionally, for potential in vivo applications, deep-tissue diffusion, off-target effects arising from protein corona formation, and potential immunotoxicity remain significant challenges.

Looking to the future, aptamer MPs could be further expanded by incorporating an upstream biochemical control layer responsive to endogenous signals, ranging from small molecules (e.g., ions, ATP), to proteins (e.g., cytokines), to even specific cell types[71]. Acting as a molecular bridge, aptamer MPs can transduce force into the exposure of cryptic ssDNA sites, which in turn can initiate localized signal amplification, recruit biomolecular effectors, modulate synthetic receptors, and ultimately trigger feedback-regulated mechanical circuits within or across cells. We envision that such aptamer-based, programmable, and feedback-coupled mechano-adaptive systems could accelerate the development of cell-autonomous behaviors, guide adaptive morphogenetic processes, and provide a foundation for future advances in synthetic biology and tissue engineering.

## Methods

### Materials
Hydrogen peroxide (35 %), sulfuric acid (95 %), (3-Aminopropyl)triethoxysilane (APTES), phosphate buffered saline (PBS), and ethanol were purchased from Carl Roth. Biotin-PEG-NHS 5 K (HE041023-5K) was purchased from Biopharma. 5(6)-(Biotinamidohexanoylamido) pentylthioureidylfluorescein was purchased from BLDpharm. Cyclo[Arg-Gly-Asp-d-Phe-Lys(PEG-PEG-azide)] (RGD-3759-PI), and Cyclo[Arg-Gly-Asp-D-Phe-Lys(Biotin)] (cRGD-Biotin, PCI-3895-PI) were purchased from Biosynth. 25 mm × 75 mm glass coverslips (10812) and sticky slide 8 well (80828) were purchased from ibidi. All oligonucleotides were purchased from Biomers. 1,2-Dioleoyl-sn-glycero-3-phosphocholine (DOPC), 5-(N-ethyl-N-isopropyl) amiloride, chlorpromazine hydrochloride, Methyl-β-cyclodextrin, LY 294002, Cytochalasin D, and Blebbistatin were purchased from Sigma-Aldrich. Streptavidin (434302), Alexa Fluor Plus 405 Phalloidin (A30104), CellTrace violet cell profileration kit (C34571), CellTrace far red cell proliferation kit (C34564), bovine albumin fraction V (BSA, 7.5% solution), paraformaldehyde solution (4% in PBS),anti-PTK7 monoclonal antibody (MA5-25774, OTI2E7), Alexa Fluor 488 anti-CD227 (Mucin-1) monoclonal antibody (53-9893-82, SM3), Alexa Fluor 488 anti-CD326 (EpCAM) monoclonal antibody (53-8326-42, MH99), Alexa Fluor 488 goat anti-mouse IgG (H + L) cross-adsorbed secondary antibody (A-11001), Alexa Fluor™ 647 goat anti-mouse IgG (H + L) highly cross-adsorbed secondary antibody (A-21236), Texas Red™ 1,2-Dihexadecanoyl-sn-Glycerin-3-Phosphoethanolamin, Triethylammoniumsalz (Texas Red™ DHPE), and Oregon Green™ 488 1,2-Dihexadecanoyl-sn-Glycerin-3-Phosphoethanolamin (Oregon Green™ 488 DHPE) were purchased from Thermo Fisher Scientific. N-WASP antibody (C-1, sc-271484) was purchased from Santa Cruz. RNase H was purchased from New England Biolabs. Actin protein (> 95% pure, rabbit skeletal muscle) was purchased from Cytoskeleton, Inc. All buffers were prepared with nuclease free water. All chemicals and enzymes were used as such without any further purification.

### DNA Hybridization
DNA concentrations were determined using a DeNovix-S-06873 (DeNovix OS 0.8.1 v4.1.5) spectrophotometer. DNA oligonucleotides were hybridized at 10 μM in a TE buffer additionally containing 12.5 mM MgCl$_2$ in 200 μL PCR tubes and subsequently diluted to 125 nM. DNA oligonucleotides were heated to 95 °C and then cooled at a rate of 1.3 °C/min to 25 °C.

### Surface preparation
Surface preparation method was modified from previously published protocols[21]. Briefly, glass coverslips (25 × 75 mm) were rinsed with ultrapure water and sequentially sonicated in water and ethanol for 20 min each. The coverslips were then immersed in freshly prepared piranha solution (3:1 v/v concentrated sulfuric acid (95 %) to hydrogen peroxide (30 %)) for 1 h. Caution: Piranha solution becomes very hot upon mixing, and is highly oxidizing and may explode upon contact with organic solvents. Then, the coverslips were washed 6 times with ultrapure water, followed by 4 successive ethanol washes. For amine functionalization, the coverslips were incubated in 3 % v/v APTES at room temperature for 1 h. The coverslips were then washed 6 times with ethanol, and baked at 80 °C for 20 min. Slides were then reacted with Biotin-PEG-NHS (1% w/v) for 1 h in ultrapure water. Next, the coverslips were washed 3 times with ultrapure water, dried under N$_2$ gas, and stored at −80 °C until use.

The biotinylated coverslips were adhered to sticky-slide 8-well chambers. Wells were incubated with 50 μg/mL of streptavidin for 1 h at room temperature, followed by three washes with nuclease-free water. Wells were then incubated with 200 μL of 250 nM probe solution (125 nM cRGD-biotin and 125 nM DNA probes) for 1 h at room temperature. After washing with nuclease free water, cell imaging media (1 % FBS, 1 % P/S, DMEM; see below) was added to the well followed by the addition of cells.

### Probe density calibration
**Lipid vesicles.** Lipid vesicles were prepared with 100% 1,2-Dioleoyl-sn-glycero-3-phosphocholine (DOPC) or with 99.5 mol% DOPC and 0.5 mol% Texas Red DHPE (TR-DHPE) or Oregon Green™ 488 DHPE (OG-DHPE). Lipid was dissolved in chloroform in a round-bottom glass tube and then carefully dried with a nitrogen stream while rotating the tube. The films were then placed in vacuum for 2 h for complete evaporation of the solvent and stored at −20 °C until further use. Small unilamellar vesicles (SUVs) were formed by sonicating the lipids in nanopure water, with a lipid concentration of 2 mg/mL. The dried lipids were resuspended in Nanopure water at 2 mg/ml by sonicating. To generate small unilamellar vesicles (SUVs), lipids were then extruded 20 times through a mini-extruder (Avanti Research 610000-1EA) assembled with a 0.1 μm polycarbonate membrane.

**Solution phase standard curve preparation.** To relate dye brightness, the brightness of known concentrations of labeled oligonucleotides and SUVs was measured in solution[72]. A glass bottom 96-well plate was first washed with ethanol and water and passivated with 1% BSA for 1 h to prevent any surface adsorption. Different concentrations of labeled oligonucleotides and SUVs (0-800 nM) were then added to each well, and their bulk fluorescence intensity in solution was measured with a fluorescence microscope (20x objective lens) to create a standard curve. The scaling factor $F$ factor was defined as:

$$F = \frac{S_{\text{labeled DNA}}}{S_{\text{labeled SUVs}}} \quad (1)$$

where $S_{\text{labeled DNA}}$ and $S_{\text{labeled SUVs}}$ represent the slopes of the plots of fluorescence intensity versus concentration for the labeled DNA and SUVs, respectively.

**Supported lipid bilayer preparation.** A glass-bottom 96-well plate was used for preparing supported lipid bilayers[21,34]. Each well was filled and soaked with ethanol for 15 min and rinsed with nuclease free water. Subsequently, 200 μL of 6 M NaOH solution was added to each well

for base etching at room temperature for 1 h. After washing each well with Nanopure water, lipid mixtures containing different percentage of labeled SUVs (0, 0.01, 0.05, 0.1, 0.25, 0.5 mol.%) were added at 0.5 mg/mL in PBS with 2.5 mM calcium ions (100 μL) to coat the glass surfaces for 20 min. After the lipid vesicles fused to the surfaces, the wells were rinsed with PBS and imaged with a fluorescence microscope to create a standard curve for probe density. Note that the substrate should never come in contact with air, as this will damage the supported membrane.

## Cell culture and transfection

HeLa (ACC 57) and HepG2 (ACC 180) were purchased from DSMZ. A549 (DSMZ, ACC 107) and MDA-MB-231 (DSMZ, ACC 732) were gifts from Tanja Weil laboratory at Max Planck Institute for Polymer Research. HeLa cells were cultured in Minimum Essential Medium (MEM) (10% fetal bovine serum (FBS), 1% penicillin–streptomycin (P/S)) at 37 °C with 5 % $CO_2$. A549 and MDA-MB-231 cells were cultured in Dulbecco's Modified Eagle Medium (DMEM) (10 % FBS, 1 % P/S) at 37 °C with 5 % $CO_2$. HepG2 cells were cultured in RPMI 1640 Medium (10 % FBS, 1 % P/S) at 37 °C with 5 % $CO_2$. mEGFP-paxillin WT plasmid was a gift from Michael Rosen (Addgene plasmid # 186152)[73]. Transfection was carried out using the Lipofectamine 3000 Transfection Reagent (Invitrogen, L3000015) following the manufacturer's instructions. N-WASP knockdown was carried out using human N-WASP siRNA (Santa Cruz, sc-36006) or control siRNA-A (Santa Cruz, sc-37007) with Lipofectamine RNAiMAX Transfection Reagent (Invitrogen, 13778030) following the manufacturer's instructions.

## Force imaging, image acquisition and analysis

All cells were seeded onto the MP surfaces in cell imaging medium (1% FBS, 1% P/S, DMEM) and incubated at 37 °C with 5 % CO2 for 4 h, except for Fig. 6h–i, where incubation times were 2, 4, and 7 h. TIRF and brightfield microscopy were performed on a Zeiss Elyra 7 Imaging System equipped with 405 nm, 488 nm, 561 nm, and 642 nm excitation lasers using alpha Plan-Apochromat 63x, N.A. 1.46.oil immersion, TIRF objective, pco.edge 4.2 CLHS water-cooled sCMOS cameras. Live cell imaging was conducted at 37 °C in humidified air with 5 % $CO_2$. Microscopy experiments were typically conducted using the following parameters: 405 nm laser: 0.4 % output (~0.01 mW), 488 nm laser: 0.8 % output (~0.7 mW), 561 nm laser: 0.8 % output (~0.8 mW), and 642 nm laser: 0.4 % output (~0.4 mW). Beam splitters (primary, secondary): LBF 405/488/561/642, SBS BP 490-560 + LP 640. Exposure time: 50 ms (except for Fig. 2d, RRX-AS1411 200 ms, Atto488-TB-5 200 ms; Fig. 5h, A549-Violet 200 ms; Fig. 5j, HeLa-Deep Red 100 ms, HepG2-Violet 200 ms). TIRF angle: 66° (n$_{oil}$: 1.518). All images were contrasted equally.

Background subtraction methods were adapted from previously published protocols[68]. Briefly, raw images were exported to FIJI (software) followed by selecting 3 ROIs in the fluorescent channel, next the background intensity was measured and subtracted from the images.

Immunofluorescence staining of N-WASP was imaged by confocal laser scanning microscopy (CLSM) using a Leica Stellaris 5 microscope (LasX v4.3.0.24308) with four laser lines and three HyD S detectors using plan-apochromat objectives (63×, 1.40 numerical aperture, oil immersion).

## Inhibition experiments

100 μM 5-(N-ethyl-N-isopropyl) amiloride (EIPA), 1.9 mM Methyl-β-cyclodextrin (MβCD), 28 μM chlorpromazine hydrochloride (CPZ) were used to inhibit the macropinocytosis, caveolae, and clathrin-mediated internalization, respectively. 50 μM LY 294002 was used to inhibit the PI3K. 10 μM cytochalasin D and 25 μM Blebbistatin were used to inhibit the actin polymerization and myosin ATPase, respectively.

## Immunostaining

Cells were fixed with 4 % (w/v) paraformaldehyde in PBS for 15 min at room temperature and permeabilized with 0.25 % (v/v) Triton X-100 in PBS for 10 min, followed by washes in PBS twice. Unspecific binding was blocked using 10 % normal goat serum in PBS for 30 min at room temperature. After removing blocking solution, cells were incubated with anti N-WASP primary antibody (1:25 dilution) at 4 °C overnight. After rinsing three times with PBS, cells were incubated in secondary antibody (1:500 dilution) and Hoechst 34580 (1:200 dilution) in 1% BSA for 1 h at room temperature. For actin staining, fixed cells were incubated with Alexa Fluor Plus 405 Phalloidin (1:200 dilution) in 2.5 % (w/v) BSA/PBS at room temperature for 1 h. Samples were then washed three times in PBS for 5 min each before microscopic observations.

## Flow cytometry

To assess the competitive binding of endostatin to nucleolin, $5 \times 10^5$ cells were incubated with 25 μg/mL endostatin and 2.5 μM FAM-AS1411 in 3 % (w/v) BSA/PBS at 4 °C for 60 min. Cells were washed and then resuspended in PBS before running through flow cytometer. To quantify the receptor expression levels, $5 \times 10^5$ cells were labelled with fluorescent primary antibodies (Alexa Fluor™ 488-anti-MUC1: 5 μg/mL, Alexa Fluor™ 488-anti-EpCAM: 5 μg/mL) in 3 % (w/v) BSA/PBS at 4 °C for 30 min. For labelling PTK7, cells were incubated with primary antibody anti-PTK7 (2 μg/mL) in 3 % (w/v) BSA/PBS at 4 °C for 30 min. Cells were washed three times with PBS and then incubated in Alexa Fluor™ 488-secondary antibody (5 μg/mL) in 3 % (w/v) BSA/PBS at 37 °C for 30 min. After labelling, cells were washed and resuspended in PBS. To assess the effect of EIPA on AS1411 internalization, $2 \times 10^5$ cells were seeded in standard cell medium in 12-well plates and incubated overnight at 37 °C. Cells were then incubated with 100 μM EIPA and 500 nM FAM-AS1411 in 1 % FBS/DMEM at 37 °C for 4 h. Cells were washed and then resuspended in PBS before running through flow cytometer.

Flow cytometry experiments were performed on a Novocyte Quanteon (Agilent, NovoExpress v.1.6.0) with 4 excitation lasers (violet 405 nm, blue 488 nm, yellow-green 561 nm, and red 640 nm) and 16 fluorescence detectors. Gating strategy is shown in Supplementary Fig. 3a. A total of 10,000 events were collected per sample. Flow cytometry data were analyzed using FLOWJO (v10.8.1) software.

## Statistical analysis

Each experiment was performed at least three independent times. No statistical method was used to predetermine sample size. No data were excluded from the analyses. The experiments were not randomized. The Investigators were not blinded to allocation during experiments and outcome assessment. Statistical results were analyzed using GraphPad Prism 8.0 and shown as the mean ± s.d. Comparisons between two groups were conducted using unpaired two-tailed Student's t-test (Figs. 4m–o, 5k), and among three or more groups using one-way ANOVA with Bonferroni test correction (Figs. 4h, 6b) as indicated. If a data set did not pass the normality tests, the significances were calculated with Mann-Whitney (Figs. 2e, 3g, 3i, 5b, d, i and 6b) for two-group comparisons or Kruskal-Wallis corrected with Dunn's test (Figs. 2b–c, 3c–e, 4b, 4f, and 5f) for multiple groups. For all analyses, *$P < 0.05$, **$P < 0.01$, ***$P < 0.001$, ****$P < 0.0001$ were considered significant. $P > 0.05$ was considered statistical non-significance.

## Reporting summary

Further information on research design is available in the Nature Portfolio Reporting Summary linked to this article.

# Data availability

Source data are provided with this paper.

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

## Acknowledgements

We would like to thank Dr. Siyu Song and Dr. Miao Xie for their helpful discussion about cell experiments. We thank Dr. Marcos Masukawa for his help with the preparation of supported lipid bilayers. We acknowledge IMB Flow Cytometry Core Facility for the flow cytometry data acquisition. T. X. acknowledges support from the Max Planck Graduate Center with the Johannes Gutenberg University of Mainz (MPGC). A.W. acknowledges funding from the European Research Council (ERC) under the European Union's Horizon 2020 research and innovation program (grant no. M3ALI: 101001638), a Gutenberg Research Professorship and a Max Planck Fellowship underpinning his Life-Like Materials Program, and from the CoM2Life start-up funds of JGU. The Zeiss Elyra 7 was funded by the Gutenberg Research College and Deutsche Forschungsgemeinschaft (DFG; grant no. 497845157).

## Author contributions

T.X. and A.W. conceived the project. T.X. designed, performed all the experiments, and conducted data analysis. S.S. helped with the surface preparation and probe immobilization. C.D. helped with TIRF measurement. T.X., S.S., C.D., and A.W. discussed the data. T.X. and A.W. wrote the manuscript with input from all authors. A.W. supervised the project.

## Funding

## Competing interests

The authors declare no competing interests.
