## [Peer Review File · Nature Communications]

Synthetic Aptamer Mechanoreceptors Enable Cell-Specific Force Sensing and Temporal Control via DNA Circuits

Corresponding Author: Professor Andreas Walther

Version 0:

Reviewer comments:

Reviewer #1

(Remarks to the Author)

This is an elegant study, the authors used aptamers to replace traditional protein/peptide ligands conjugated to DNA force probes. This allowed them to target proteins beyond classical mechanoreceptors with existing aptamers. Additionally, the construction of the aptamer DNA force probes became significantly simplified, requiring no conjugation to proteins. This advancement broadens the receptors that can be targeted by force probes and giving researchers the ability to control the interactions using established DNA reactions.

Specifically, the study showed several examples applications. First, they targeted nucleolin to establish the concept with both unzipping and shearing DNA geometries. They showed that both Blebbistatin and Cytochalasin D reduced the nucleolin-AS1411 tension signal. The authors elucidated the tension mechanisms using different drugs (CPZ, MbCD, and EIPA) and showed specific signal reduction due to EIPA but not the other 2, linking the source of tension to macropinocytosis. Additionally, the demonstration of ring-like structures using PTK7 force probes further showed PTK7 tension signal is directly related to macropinocytosis. The authors showed that different cells with and without certain receptor expression can be detected by the presence and absence of aptamer tension signals. Lastly, they demonstrated using various reaction schemes (e.g., DNA strand displacement, RNaseH) to reversibly activate and deactivate the aptamer force probes.

Overall, the paper did an excellent job demonstrating the utility and advantages of aptamer in DNA tension probes. Both the text and figure are clearly presented, I believe this is an important step in the evolution of DNA-based force probes.

I have a few comments that I hope the authors could address to strengthen the manuscript. Two major comments first:

1. An important aspect of the manuscript is to enable studying tension from non-canonical mechanoreceptors. While I believe this is of significant interest, the authors should elaborate on how investigations using non-native interactions (i.e. with aptamers) yield biological relevance for native interactions. For instance, if a receptor's native ligand binds weakly and only allow low force transmission, a high affinity aptamer would report high force transmission, particularly if the receptor is tightly associated with... say, the focal adhesion complex. The tension signal here would report the capacity of tension generation, rather than the receptor's functional tension with its native ligands. I think this distinction could be discussed in more detail, and claims related to the utilities using aptamers could be tightened accordingly. That said, I also I very much appreciate the accuracy of statements such as "the AS1411-nucleolin couple transmits substantially lower force than RGD-integrin engagement.", which highlights the context of the comparison is with the specific interaction pairs.

2. The data on cell-type specific mechanoresponses is clean. But it is also kind of obvious that cells with and without receptors would and wouldn't produce tension signals. By testing simple presence or absence, this section seems more like a control for aptamer specificity. The wording of claim for being able to discern "cell-type specific mechanoresponse" would suggest that given the same existing mechanoreceptor on different cell types, one can differentiate tension signals (e.g. magnitude, spatial patterns, etc) as a result of their different cellular physiology. The "mechano-response" should produce additional details that examining receptor expression level alone (e.g. with immunofluorescence with FC) cannot reveal. I would strongly encourage an additional experiment with 2 cell types that both express say, PTK7, but show that they produce different tension signal – that would be very nice! In the absence of such experiment, the claim would be more limited, and the text should be tightened accordingly.

Minor comments:

3. Could the authors elaborate why in the co-culture settings (Fig 5i,k), the differences between the mean RRX signals aren't as tight as in separate monoculture (Fig 5b,d)? Additionally, could you also elaborate on the variability in RRX intensity? Esp for cells that do not express the receptor, some show significant tension signal (Fig 5k) in comparison to the MUC1(+) cells. The authors should also refer to their flow cytometry data in Supp. Fig 9 in this discussion (maybe I missed it).

4. The manuscript presents mean and integrated RFX intensity in various places, how should we interpret the difference between them (e.g. Fig 3c,d)? And why is integrated intensity included in some places but not others (Fig 5b,d,f)? If it's not adding useful information, I would suggest removing integrated intensity from Fig 5b,d,f, to keep consistency throughout.
5. Nucleases in the live-cell media (either from serum or cell-release) have been problematic for long-term studies using DNA-based probes. Have the authors observed any problems with DNA degradation and non-specific fluorescence signals in this system? I think a short discussion on protecting native aptamers from such degradation could be useful to elevate its impact. There have been several strategies using PNA (<https://www.sciencedirect.com/science/article/pii/S095656631931036X>), your L-DNA work (<https://onlinelibrary.wiley.com/doi/10.1002/anie.202413983>), PS mod (<https://pubs.acs.org/doi/10.1021/acsami.3c04826>), and probably more relevant for protecting unmodified nucleic acid suitable for aptamers with a decoy DNA (<https://onlinelibrary.wiley.com/doi/10.1002/anie.202506590>).
6. Maybe I missed it, could the authors indicate the time frame in the RHaseH study (Fig 6e,f)? Could you also comment on the reaction rate using RHaseH or strand-displacement (using a reasonable amount of reagent)? Which one is faster?
7. There are some existing work on using strand displacement to control junctional tension such as (<https://pubs.acs.org/doi/10.1021/acsabm.4c00142>) I think a short discussion of existing literature on force mediation/modulation using DNA reactions could strengthen the manuscript and help highlight differences and your advantages – modulating function without destroying the probe.

Reviewer #2

(Remarks to the Author)

In this manuscript, Xu et al. present a modular all-DNA aptamer-based mechanosensing platform. The authors integrate receptor-specific recognition via aptamers with the tunable force sensitivity of DNA mechanoprobes. A key strength of the study is the demonstration of dual selectivity, discriminating cellular forces based on both cell type and discrete force thresholds. The investigation of non-canonical mechanoreceptors such as nucleolin and PTK7 provides valuable new insights into how cells mechanically interact with their microenvironment. Moreover, the integration of the platform with DNA reaction networks to achieve programmable temporal control represents a significant methodological advance for synthetic mechanobiology. Overall, this work is well executed and presents a notable contribution to the field. However, several mechanistic claims and aspects of validation should be further substantiated to strengthen the rigor and overall impact of the work. Therefore, I recommend minor revision prior to acceptance for publication in Nature Communications.

Major concerns and suggestions

1. The claim that nucleolin-mediated forces are “relayed via focal adhesions” is central to the mechanistic interpretation. The current evidence—paxillin colocalization and sensitivity to myosin II inhibition—supports spatial and functional correlation but not direct physical linkage. An alternative explanation, that nucleolin and integrins are independently coupled to the same actomyosin network, cannot be excluded. It is recommended that the authors either moderate this claim or propose additional experiments (e.g., FRET, co-immunoprecipitation) to establish a direct link.
2. In Figure 4, the mechanistic investigation relies exclusively on small-molecule inhibitors, which are prone to off-target effects. The distinction between myosin-driven (nucleolin) and actin-polymerization-driven (PTK7) forces would be more convincing if validated with orthogonal genetic perturbations, such as shRNA knockdown of N-WASP.
3. A critical control is missing to validate the claim that the aptamer-based force sensing is functionally independent of the RGD-mediated adhesion used throughout the study. To robustly prove this independence, the assay should also be performed on a surface functionalized with the aptamer MPs alone (i.e., without co-immobilized RGD). This would directly test whether baseline integrin adhesion is a prerequisite for the aptamer probes to function, making the conclusion in Figure 2c more convincing.
4. The actual surface density and stoichiometry of the immobilized RGD and MP molecules were not quantified. These parameters are known to significantly influence cell adhesion and force generation. Providing this characterization would greatly enhance the rigor and reproducibility of the platform.
5. The SDR module is currently demonstrated with only a single block–activate cycle, which is not sufficient to establish reversibility. It is recommended that the authors present at least two or more consecutive suppression–recovery cycles. In addition, the RNase H-based “timer” shows tunable activation rates but not a clearly defined lag phase. Including earlier and more frequent time points would help demonstrate a controllable delay before signal onset, and a simple kinetic model correlating decoy concentration with delay would further substantiate the claim of programmability.
6. The statement that aptamer MPs exhibit “enhanced force selectivity” compared to RGD MPs may be misleading. A more accurate interpretation is that AS1411–nucleolin interactions occur within a lower physiological force regime (consistently below 54 pN), while RGD–integrin forces readily exceed this threshold. This distinction reflects biological force domains rather than an intrinsic property of the probe.
7. Regarding the citation of Yang et al. (Ref. 20), the manuscript currently describes this study as “force being sensed using a DNA MP”, which may not fully reflect its contribution. Specifically, Yang et al. provided the proof-of-concept that a DNA aptamer, taking CI-M6RP-targeting aptamer as the example, can function as a force-reception module to sense weak endocytic forces with fluorescent signal readout and then translate these force cues into intracellular signals via DNA dynamic reactions. Thus, a more balanced description should acknowledge this advance in the revision, and it is recommended to add a brief discussion clarifying how the present work extends this concept.

Minors

1. Use more precise terminology such as “receptor-low” or “low-expressing” instead of “receptor-negative,” as the supplementary flow cytometry data indicate non-zero expression.

2. Clarify in the text or figure legend for Figure 3 why both “mean” and “integrated” intensity are presented, and how each metric provides distinct insights.
3. In Figure 6, the restored force signal distribution appears inconsistent across conditions (e.g., recovery mainly at the cell edge vs. both edge and center). The authors should clarify whether these differences reflect mechanistic variations or are attributable to experimental variability.
4. For PTK7-mediated forces, it remains unclear whether the signal reflects protrusive pushing or contractile pulling. The schematic suggests pulling, whereas the text emphasizes polymerization-driven pushing. This ambiguity should be explicitly acknowledged and discussed.
5. In the discussion, briefly acknowledge the limitations of DRN-controlled systems for potential in vivo applications, including delivery challenges, nuclease stability, and cytotoxicity.
6. While the manuscript identifies novel force pathways, the discussion could be enriched by addressing their biological significance. Why do cells exert forces through these non-canonical receptors? Furthermore, since detailed analysis was limited to nucleolin and PTK7, it would be valuable to comment on the potential generalizability to other possible aptamer–receptor pairs with biological functions.

Version 1:

Reviewer comments:

Reviewer #1

(Remarks to the Author)

I appreciate the careful revision from the authors. Excellent work!

I have reviewed all the responses to my previous questions and the revised manuscript. I believe the manuscript is suitable for publication in Nat. Comms. without further revision.

Reviewer #2

(Remarks to the Author)

The authors have adequately addressed the concerns raised in my previous review by providing substantial additional experimental data and analyses. The revised manuscript presents strengthened mechanistic support for the main conclusions. Overall, the data support the claims, the methodology is sound, and the current version meets the standards for publication in Nature Communications. I therefore support acceptance of the manuscript in its present form.

Reviewers comments:

Reviewer #1 (Remarks to the Author)

This is an elegant study, the authors used aptamers to replace traditional protein/peptide ligands conjugated to DNA force probes. This allowed them to target proteins beyond classical mechanoreceptors with existing aptamers. Additionally, the construction of the aptamer DNA force probes became significantly simplified, requiring no conjugation to proteins. This advancement broadens the receptors that can be targeted by force probes and giving researchers the ability to control the interactions using established DNA reactions. Specifically, the study showed several examples applications. First, they targeted nucleolin to establish the concept with both unzipping and shearing DNA geometries. They showed that both Blebbistatin and Cytochalasin D reduced the nucleolin-AS1411 tension signal. The authors elucidated the tension mechanisms using different drugs (CPZ, MbCD, and EIPA) and showed specific signal reduction due to EIPA but not the other 2, linking the source of tension to macropinocytosis. Additionally, the demonstration of ring-like structures using PTK7 force probes further showed PTK7 tension signal is directly related to macropinocytosis. The authors showed that different cells with and without certain receptor expression can be detected by the presence and absence of aptamer tension signals. Lastly, they demonstrated using various reaction schemes (e.g., DNA strand displacement, RNaseH) to reversibly activate and deactivate the aptamer force probes. Overall, the paper did an excellent job demonstrating the utility and advantages of aptamer in DNA tension probes. Both the text and figure are clearly presented, I believe this is an important step in the evolution of DNA-based force probes.

I have a few comments that I hope the authors could address to strengthen the manuscript. Two major comments first:

Thank you for your very positive evaluation of our work.

Comment 1-1. An important aspect of the manuscript is to enable studying tension from non-canonical mechanoreceptors. While I believe this is of significant interest, the authors should elaborate on how investigations using non-native interactions (i.e. with aptamers) yield biological relevance for native interactions. For instance, if a receptor's native ligand binds weakly and only allow low force transmission, a high affinity aptamer would report high force transmission, particularly if the receptor is tightly associated with... say, the focal adhesion complex. The tension signal here would report the capacity of tension generation, rather than the receptor's functional tension with its native ligands. I think this distinction could be discussed in more detail, and claims related to the utilities using aptamers could be tightened accordingly. That said, I also I very much appreciate the accuracy of statements such as "the AS1411-nucleolin couple transmits substantially lower force than RGD-integrin engagement.", which highlights the context of the comparison is with the specific interaction pairs.

Response: We thank the reviewer for this insightful comment. While aptamer-based platforms represent a promising and accessible tool for the pre-screening of non-canonical mechanoreceptors, we fully agree that extrapolating mechanoevents detected using aptamer-receptor pairs to native ligand-receptor interactions must be done with caution. Several key factors require careful consideration:

1. **Immobilization state of the ligand** is critical, as it determines the magnitude of the reaction force that can be supported. In our system, aptamer probes are immobilized via biotin-streptavidin interaction, which provides sufficient reaction force to resist cellular pulling. Mobile or weakly immobilized ligands may lead to inefficient force transmission.

2. Binding affinity and bond mechanics. As the reviewer noted, differences in binding affinity may affect force transmission. Overly stable aptamer-receptor interactions may lead to an overestimation of the force transmitted under native conditions. In addition, while aptamer-receptor interactions typically exhibit slip bond behavior, native ligand-receptor pairs, especially classical mechanoreceptors such as RGD-integrin, pMHC-TCR, etc., may involve complex catch bonds. This difference in bond lifetime under load may significantly change the mechanical behaviors.

3. Multivalency and cooperative effects. Synthetic aptamers often lack the multi-domain architecture of native ligands (e.g., RGD and LDV motifs present in fibronectin).¹ These differences may lead to variations in inside-out force transmission, potentially influencing unknown outside-in mechanotransduction pathways and thus affecting result interpretation. These limitations are inherent to studies using simplified synthetic ligands.

Regarding the effect of binding affinity as the reviewer mentioned, the AS1411 aptamer exhibits a K_d of 69.1 nM for nucleolin, whereas its native ligand endostatin possesses a K_d of 23.2 nM. Since the endostatin's affinity is approximately 3-fold higher, it is highly possible that our platform underestimates the native capacity for tension generation. Despite this, we still observe the mechanosignals, suggesting such affinity is sufficient for partial force transmission. The ability of nucleolin to transmit force is indicated by existing literature: (i) Nucleolin mediates the internalization of endostatin in endothelial cells by forming a receptor complex with integrin $\alpha_5\beta_1$,^{2,3} a process we consider a mechanoevent, which cell survival and proliferation. (ii) Nucleolin is known to associate with cytoskeletal components, including integrins,^{2,4,5} focal adhesion proteins,⁶ and MYH9,⁷ which provides a structural basis for its role in force transmission. Notably, endostatin is derived from the C-terminal non-collagenous (NC1) domain of type XVIII collagen in the extracellular matrix and is generated by MMP-mediated cleavage.⁸ Whether nucleolin can transmit force through this precursor protein in the native ECM context, and whether endostatin-associated biological functions are linked to the non-canonical mechanical properties of nucleolin, remain important questions for future investigation. Addressing these questions may reveal how cryptic motifs within the ECM regulate cellular homeostasis through non-canonical mechanotransduction pathways.

Conversely, as the reviewer noted, if an aptamer possesses a higher affinity than the native ligand, the resulting over-stabilized mechanical linkage might overestimate natural tension capacity. In such cases, beyond serving as a detection probe, the aptamer MPs may be better viewed as functional actuators. This relative independence and orthogonality make it well suited for integration with DNA nanotechnology and artificial extracellular matrix materials for tissue engineering purposes to construct artificial signaling pathways for controlled modulation of cellular behaviors.

We have revised the related discussion in the text:

The identification of noncanonical forces highlight the importance of revisiting the biological functions of both these aptamers and receptors from a mechanobiological perspective. Determining whether these receptors are directly coupled to classical adhesion structures or are independently linked to the same actomyosin network via intermediary components will be essential for uncovering more general principles. Whether similar interactions occur under native conditions remains to be elucidated. A prime candidate is the precursor of the nucleolin ligand endostatin, located in the NC1 domain of type XVIII collagen,⁶⁰ which may serve as a native anchor for nucleolin despite potential differences in binding affinity. Addressing these questions could reveal how cryptic motifs within the ECM regulate cellular homeostasis through non-canonical mechanotransduction pathways. Meanwhile, caution is required when extrapolating from aptamer-receptor pairs to native ligand-receptor interaction, particularly regarding the immobilization state of the ligand, binding

affinity and bond mechanics, and the multivalency and cooperative effects inherent to native ligands. For example, the AS1411 aptamer binds nucleolin with a K_d of 69.1 nM,²⁸ which is lower than that of its native ligand endostatin (23.2 nM),⁶¹ suggesting that our measurements may slightly underestimate the native capacity for tension generation.

Comment 1-2. The data on cell-type specific mechanoresponses is clean. But it is also kind of obvious that cells with and without receptors would and wouldn't produce tension signals. By testing simple presence or absence, this section seems more like a control for aptamer specificity. The wording of claim for being able to discern "cell-type specific mechanoresponse" would suggest that given the same existing mechanoreceptor on different cell types, one can differentiate tension signals (e.g. magnitude, spatial patterns, etc) as a result of their different cellular physiology. The "mechano-response" should produce additional details that examining receptor expression level alone (e.g. with immunofluorescence with FC) cannot reveal. I would strongly encourage an additional experiment with 2 cell types that both express say, PTK7, but show that they produce different tension signal – that would be very nice! In the absence of such experiment, the claim would be more limited, and the text should be tightened accordingly.

Response: Thanks for the suggestion. We indeed identified two different cell lines that both express the same receptor but exhibit different force patterns. Compared with PTK7 high-expressing HepG2 and PTK7 low-expressing A549 cells, flow cytometry data demonstrate that HeLa cells display an intermediate expression level of PTK7 (Supplementary Fig. 11e,f). Interestingly, mechanosensing using Sgc8 MPs reveals not only an expression-level-dependent signal magnitude but also different force patterns. Unlike the characteristic ring-like mechanosignals observed in HepG2 cells, HeLa cells exhibit a stripe-like force pattern (Fig. 5e,f). These results indicate that even when the same receptor is expressed, different cell types may transmit forces through that receptor via distinct pathways, and thus leads to different mechanoresponses in both signal magnitude and force patterns. We added this new data to Fig. 5e,f and Supplementary Fig. 11e,f and discussed it in the text.

Fig. 5 e Representative brightfield and fluorescence images of HepG2 (PTK7-high-expressing), HeLa (PTK7-intermediate-expressing) and A549 (PTK7-low-expressing) cells on Sgc8 MP surfaces. **f** Corresponding quantification of mean and integrated fluorescence intensity per cell of HepG2, HeLa and A549 cells on Sgc8 MP surfaces. n = 36, 31, 27 cells from 3 replicates. Statistics: Kruskal-Wallis test with Dunn's multiple comparisons.

Supplementary Figure 11. **e** Representative flow cytometry histogram of HepG2, HeLa, and A549 cells stained with anti-PTK7 primary antibody followed by Alexa Fluor™ 488-conjugated secondary antibody. Mock controls represent unstained cells, indicating cellular autofluorescence. **f** Quantification of PTK7 expression levels on HepG2, HeLa, and A549 cells based on median fluorescence intensity. $n = 3$ independent experiments. Statistics: One-way ANOVA with Bonferroni post-hoc tests.

A discussion for this result has been added to the main text:

Interestingly, beyond expression level-dependent signal magnitude, different cell types expressing the same receptor can exhibit distinct, cell-type-dependent force patterns. HeLa cells express PTK7 at an intermediate level (Supplementary Fig. 11e,f). Mechanosensing using Sgc8 MPs reveals a stripe-like force pattern in contrast to the characteristic ring-like mechanosignals observed in HepG2 cells (Fig. 5e, f). These results indicate that different cell types may transmit forces through the same receptor via distinct pathways, a phenomenon that cannot be inferred from receptor expression levels alone.

Minor comments:

Comment 1-3. Could the authors elaborate why in the co-culture settings (Fig 5i,k), the differences between the mean RRX signals aren't as tight as in separate monoculture (Fig 5b,d)? Additionally, could you also elaborate on the variability in RRX intensity? Esp for cells that do not express the receptor, some show significant tension signal (Fig 5k) in comparison to the MUC1(+) cells. The authors should also refer to their flow cytometry data in Supp. Fig 9 in this discussion (maybe I missed it).

Response: Thank you for the question. We are sorry that our original terminology, “receptor-negative” and “receptor-positive”, are not precise enough. As noted by Reviewer#2 (question 2-8), and shown in Supplementary Fig. 11 (original Supp. Fig 9), A549 and HepG2 cells do express basal levels of PTK7 and MUC1 receptors, respectively, meaning they are not truly “negative” cell lines. The mechanosignal distribution is strongly influenced by the overlap in receptor expression level between these populations. Specifically, the higher mechanosignals observed in “low-expressing” groups may come from those few cells that happen to have a high expression of receptors. Consequently, the breadth of the signal distribution depends on whether these overlapped cells are captured during sampling. This makes the results sensitive to sample size, and batch-to-batch variations in expression. While the transition from monoculture to co-culture conditions may have some effect, we do not wish to speculate excessively on this behavior beyond what is directly supported by the data. To reflect this, we have updated our terminology to “low-expressing” and “high-expressing”

throughout the manuscript. We also discussed the original Supp. Fig 9 (now Supplementary Fig. 11) in the text.

Comment 1-4. The manuscript presents mean and integrated RRX intensity in various places, how should we interpret the difference between them (e.g. Fig 3c,d)? And why is integrated intensity included in some places but not others (Fig 5b,d,f)? If it's not adding useful information, I would suggest removing integrated intensity from Fig 5b,d,f, to keep consistency throughout.

Response: Thank you for the question. Our analysis follows the well-established methodology for RGD-based MPs developed by the Salaita group.^{9,10} We distinguish between mean intensity, which represents the average mechanosignal per unit area within the cell footprint, and integrated intensity, which reflects the cumulative mechanosignal across the entire footprint and is thus inherently dependent on cell area.¹¹

Under idealized conditions where cell spreading remains invariant across different probe surfaces, the trends for mean and integrated intensities should converge: higher tension tolerance T_{tol} probes offer greater resistance to rupture, theoretically yielding lower values for both metrics. Previous literatures confirm that mean intensity captures these T_{tol} -dependent shifts.¹¹ Cells on 12 pN RGD-MP surfaces typically exhibit robust, uniform rupture, whereas cells on 54 pN RGD-MP surfaces yield a streak pattern, resulting in a lower mean intensity.

However, the magnitude of integrated intensity is often inconsistent across studies because it is cross-regulated by both the intrinsic T_{tol} of the probe and the cell spreading area.^{10,11} The latter is highly sensitive to surface chemistry or probe geometries themselves. In the absence of exogenous adhesion enhancers, cells on 12 pN RGD-MP surfaces often fail to achieve adhesion maturation as the reaction force required typically exceeds 43 pN. This results in restricted cell area, in which the integrated intensity may paradoxically appear lower than that of cells on 54 pN surfaces. Conversely, when spreading is promoted by adhesion enhancers (e.g., fibronectin, poly-L-lysine) or mixed- T_{tol} surfaces, cells on 12 pN surfaces exhibit greater sensitivity to these cues to increase cell area, while 54 pN surfaces inherently provide sufficient mechanical resistance to support adhesion. As a result, when cells are equally well-spread on both surfaces, the 12 pN surfaces yield a higher integrated fluorescence, reflecting the lower threshold for probe rupture.

Given these complexities, we believe it is essential to present integrated intensity whenever variations in cell area cannot be neglected. This is particularly relevant when comparing different probes that may alter spreading behavior (Fig. 3c, d) or when examining different cell lines with distinct basal morphologies (Fig. 5b, d, f), such as the smaller MDA-MB-231 cells versus the larger HeLa cells. Furthermore, the integrated intensity provides the most accurate representation of the total stimulatory input per individual cell, which is the primary driver to design downstream reactions.

A discussion for this result has been added to the main text:

The integrated intensity reflects the cumulative mechanosignal across the entire cell footprint and is thus not only regulated by intrinsic T_{tol} of the probe but inherently dependent on cell area.¹⁹

In addition, this discussion provided now in the answer letter (that is available to readers in Nat. Commun.) provides additional clarification.

We also updated the figure captions:

Fig.3 c-e Quantification of (c) mean fluorescence intensity per cell, (d) integrated fluorescence intensity per cell, and (e) cell spreading area for HeLa cells on each MP type and geometry. n = 49, 47, 45, 36 (left to right)

cells from 3 replicates. Statistics: Kruskal-Wallis test with Dunn's multiple comparisons. Note that both mean and integrated intensity are shown because mean intensity mainly captures the T_{tot} -dependent shifts while integrated intensity is cross-regulated by both the intrinsic T_{tot} of the probe and the cell spreading area.

Fig.5 b Corresponding quantification of mean and integrated fluorescence intensity per cell of HeLa and HepG2 cells on MUC1 S2.2 MP surfaces. $n = 31, 27$ cells from 3 replicates. Statistics: Two-tailed, Mann-Whitney test. Note that both mean and integrated intensity are shown here because different cell lines may have distinct basal morphologies with different cell area.

Comment 1-5. Nucleases in the live-cell media (either from serum or cell-release) have been problematic for long-term studies using DNA-based probes. Have the authors observed any problems with DNA degradation and non-specific fluorescence signals in this system? I think a short discussion on protecting native aptamers from such degradation could be useful to elevate its impact. There have been several strategies using PNA (<https://www.sciencedirect.com/science/article/pii/S095656631931036X>), your L-DNA work (<https://onlinelibrary.wiley.com/doi/10.1002/anie.202413983>), PS mod (<https://pubs.acs.org/doi/10.1021/acscami.3c04826>), and probably more relevant for protecting unmodified nucleic acid suitable for aptamers with a decoy DNA (<https://onlinelibrary.wiley.com/doi/10.1002/anie.202506590>).

Response: Thank you for the question and providing these useful papers for reference. In our study, we adopted a protocol like those reported in the literature,^{11,12} using low-serum media (1% FBS/DMEM) to mitigate the effects of exogenous nucleases. This approach with appropriate background subtraction provides reliable results during short-term observations. Regarding endogenous nucleases secreted by cells, Fig. 2b demonstrates that the signal from the "null probe" accounts for both nuclease-mediated degradation and non-specific interactions (such as dye-membrane interactions). Compared to the robust mechanosignals observed for AS1411 MPs, these nonspecific signal levels remain within an acceptable range.

But we fully agree that additional nucleic acid protection strategies are essential for long-term practical applications. In our RNA-RNase H kinetic assays extending up to 7 h, we introduced actin as a competitive nuclease inhibitor to preserve the structural integrity of DNA MPs.¹³ While actin offers effective protection, its high cost limits broader application. Moving forward, we believe that integrating a combination of the recommended strategies will significantly enhance the stability of aptamer-based platforms. Regarding the duplex-based force probes, incorporating peptide nucleic acid or phosphorothioate modifications into the double-stranded regions or directly using L-DNA could indeed provide resistance against nuclease-mediated degradation. But the situation differs for single-stranded aptamers, which are inherently more susceptible to degradation. Any chemical modification intended to increase stability must be carefully balanced against potential losses in targeting affinity. In this context, we believe that employing decoy DNA is a better solution, as it provides an optimal balance between protective efficacy and cost-effectiveness. Please also note that the sequence-to-force relationships (well established for pristine D-DNA and L-DNA) would also need to be re-established by single molecule force measurements when applying modifications such as PDNA or locked nucleic acids.

We cited these references and discussed in the main text:

However, long-term application in these environments still faces stability challenges. While modifications like peptide nucleic acids,⁶³ phosphorothioate linkages,⁶⁴ or our recent L-DNA strategy protect the MP structure,⁶⁵ single-stranded aptamers remain inherently degradation-prone. Balancing their structural integrity with

binding affinity is critical. In this context, using nuclease inhibitors such as actin or decoy DNA may be more effective.^{66,67}

Comment 1-6. Maybe I missed it, could the authors indicate the time frame in the RNaseH study (Fig 6e,f)? Could you also comment on the reaction rate using RNaseH or strand-displacement (using a reasonable amount of reagent)? Which one is faster?

Response: Thank you for the question. (1) Time frame: We immobilized the RNA blocker-locked AS1411 probes on the surface, seeded the HeLa cells and added RNase H at varying concentration (0-500 U/mL) simultaneously. Cells were imaged after 4 h of incubation. We updated these experimental details to the figure caption.

(2) Speed of reactions (new experiments): We monitored the reaction kinetic of strand displacement reaction (SDR) and RNA-RNase H modules to assess reaction rates. A FRET pair (Atto488/BHQ1) was incorporated into both the AS1411 strand and the DNA/RNA blocker so that we can quantify the ratio of AS1411 aptamer reconfiguration. We selected RNase H concentrations of 10 U/mL and 100 U/mL for comparison with SDR using 200 nM DNA activator, as no obvious signal recovery was observed at lower RNase H concentration, whereas signal saturation occurred at higher RNase H concentration (Fig.6e). The reaction rate of SDR with 200 nM DNA activator ($2.14 \times 10^{-2} \text{ min}^{-1}$) lies between those observed for RNase H at 10 U/mL ($0.71 \times 10^{-2} \text{ min}^{-1}$) and 100 U/mL ($5.26 \times 10^{-2} \text{ min}^{-1}$). We added this new data to Supplementary Fig. 12d, Supplementary Fig. 13a, and Supplementary Table 3, and discussed it in the text.

Combined representation of Supplementary Figure 12d and 13a. Plot of normalized surface atto488 intensity versus time to compare the reconfiguration rate of AS1411 aptamer in SDR and RNA-RNase H modules. For the SDR module, surface-immobilized AS1411 MPs are initially blocked with DNA blocker and subsequently reconfigured by adding 200 nM DNA activator. For the RNA-RNase H module, surface-immobilized AS1411 MPs are blocked with RNA blocker and degraded by RNase H at different concentrations (10 or 100 U/mL). At each reaction time point, the surface Atto488 intensity is normalized to that of unblocked AS1411 MPs labeled with atto488, which serves as the positive control. Reaction medium: 1 % FBS, 1 % P/S, 100 $\mu\text{g}/\text{mL}$ actin protein, DMEM. n = 15 from 3 replicates.

Supplementary Table 3. Reaction rate constant of SDR and RNA-RNase H modules. To understand the reconfiguration efficiency and tunability of different DNRs, we quantified the surface reconfiguration kinetics. The data were analyzed assuming pseudo-first-order kinetics and fitted using one phase exponential decay function. For the SDR module, reactivation with 200 nM DNA activator yields a reconfiguration rate of $2.14 \times 10^{-2} \text{ min}^{-1}$. The RNA-RNase H module enables degradation rate tuning by adjusting the initial RNase H concentration, resulting in accelerated (100 U/mL, $5.26 \times 10^{-2} \text{ min}^{-1}$) or decelerated (10 U/mL, $0.71 \times 10^{-2} \text{ min}^{-1}$) reconfiguration kinetics. Moreover, at a fixed RNase H concentration (10 U/mL), the RNA-RNase H module allows the introduction of non-enzymatic RNA/DNA decoys, providing an additional orthogonal layer of regulation.

	$k (\times 10^{-2} \text{ min}^{-1})$	R^2
SDR 200 nM DNA activator	2.14	0.9926

RNase H 100U/mL	5.26	0.9925
RNase H 10U/mL	0.71	0.9842
RNase H 10U/mL+1 nM rdm RNA/DNA decoy	0.73	0.9782
RNase H 10U/mL+10 nM rdm RNA/DNA decoy	0.42	0.9603
RNase H 10U/mL+100 nM rdm RNA/DNA decoy	0.11	0.9831

A discussion for this result has been added to the main text:

Subsequent medium exchange with the equimolar DNA activator restores ~ 50 % of the signal within 30 min with a reaction rate of $2.14 \times 10^{-2} \text{ min}^{-1}$, confirming aptamer reactivation (see Methods, Supplementary Fig. 12a-d, Supplementary Table 3).

Monitoring the aptamer reconfiguration kinetics with simple fluorescence assays reveals RNase H concentration-dependent reaction rates of $0.71 \times 10^{-2} \text{ min}^{-1}$ at 10 U/mL and $5.26 \times 10^{-2} \text{ min}^{-1}$ at 100 U/mL (see Methods, Supplementary Fig. 13a, Supplementary Table 3).

Comment 1-7. There are some existing work on using strand displacement to control junctional tension such as (<https://pubs.acs.org/doi/10.1021/acscabm.4c00142>) I think a short discussion of existing literature on force mediation/modulation using DNA reactions could strengthen the manuscript and help highlight differences and your advantages – modulating function without destroying the probe.

Response: Thanks for the suggestion of more references. A discussion for has been added to the main text:

Moving forward, pioneering work using protein-ligand functionalized MPs has shown that strand displacement can control intracellular mechanics in 3D spheroids,⁶² although at the cost of ligand release. Introducing aptamer-based MPs further enables non-invasive, tunable modulation without compromising probe integrity.

Reviewer #2 (Remarks to the Author)

In this manuscript, Xu et al. present a modular all-DNA aptamer-based mechanosensing platform. The authors integrate receptor-specific recognition via aptamers with the tunable force sensitivity of DNA mechanoprobes. A key strength of the study is the demonstration of dual selectivity, discriminating cellular forces based on both cell type and discrete force thresholds. The investigation of non-canonical mechanoreceptors such as nucleolin and PTK7 provides valuable new insights into how cells mechanically interact with their microenvironment. Moreover, the integration of the platform with DNA reaction networks to achieve programmable temporal control represents a significant methodological advance for synthetic mechanobiology. Overall, this work is well executed and presents a notable contribution to the field. However, several mechanistic claims and aspects of validation should be further substantiated to strengthen the rigor and overall impact of the work. Therefore, I recommend minor revision prior to acceptance for publication in Nature Communications.

Thank you for your very positive evaluation of our work.

Major concerns and suggestions

Comment 2-1. The claim that nucleolin-mediated forces are “relayed via focal adhesions” is central to the mechanistic interpretation. The current evidence—paxillin colocalization and sensitivity to myosin II inhibition—supports spatial and functional correlation but not direct physical linkage. An alternative explanation, that nucleolin and integrins are independently coupled to the same actomyosin network, cannot be excluded. It is recommended that the authors either moderate this claim or propose additional experiments (e.g., FRET, co-immunoprecipitation) to establish a direct link.

Response: We thank the reviewer for this insightful suggestion. We agree that establishing a direct physical link between nucleolin and focal adhesions is essential for a complete molecular picture. However, fully characterizing these protein-protein interactions requires a substantial body of work, which we consider to rather warrant dedicated follow up studies.

Previous studies have reported interactions between nucleolin and multiple focal adhesion-related or cytoskeletal components, including integrins $\alpha_v\beta_3$,⁴ $\alpha_5\beta_1$,² and β_1 ,⁵ the focal adhesion protein talin,⁶ and MYH9,⁷ as identified by Co-IP or Co-IP/MS. These findings indicate that nucleolin and focal adhesion-associated proteins can coexist within the same molecular complexes; however, they also do not exclude the presence of intermediary linker proteins that may independently connect nucleolin to the complexes or actomyosin network.

As suggested by the reviewer, definitive validation of direct interactions would require approaches such as *in situ* intermolecular FRET or *in vitro* pull-down assays using purified proteins. However, both methods present significant challenges: (i) FRET require extensive protein engineering to generate fluorescent fusion constructs, which is technically demanding, and may sterically hinder the native interaction or alter the localization of nucleolin. (ii) *In vitro* pull-down using purified proteins cannot fully recapitulate the post-translational modifications or the specific membrane microenvironment that may be critical for nucleolin’s binding affinity in live cells. This is critical as nucleolin exists in both cytoplasmic and membrane-bound forms, which may exhibit distinct binding affinities for the same partner protein.

Given the complexity of high-throughput screening and the rigorous requirements of *in situ* validation, we cannot provide a definitive conclusion within the current revision timeframe. We have therefore softened our claim and acknowledged this limitation in the Discussion Section:

Determining whether these receptors are directly coupled to classical adhesion structures or are independently linked to the same actomyosin network via intermediary components will be essential for uncovering more general principles.

Comment 2-2. In Figure 4, the mechanistic investigation relies exclusively on small-molecule inhibitors, which are prone to off-target effects. The distinction between myosin-driven (nucleolin) and actin-polymerization-driven (PTK7) forces would be more convincing if validated with orthogonal genetic perturbations, such as shRNA knockdown of N-WASP.

Response: Thanks for this insightful suggestion. We used a commercially available N-WASP (human) siRNA and successfully knocked down intracellular N-WASP in both HeLa cells (nucleolin high-expressing) and HepG2 cells (PTK7 high-expressing). After detachment, these cells were seeded onto AS1411 MPs or Sgc8 MPs surface, respectively, to probe the corresponding mechanosignals.

For the AS1411-nucleolin pair, compared with cells transfected with scrambled siRNA, N-WASP knockdown HeLa cells still exhibit a stripe-like force pattern, although the signal amplitude is reduced by 27.9% (Fig. 4n, Supplementary Fig. 10a). This result indicates that force transmission through the AS1411-nucleolin pair is weakly dependent on N-WASP. In contrast, for the Sgc8-PTK7 pair, N-WASP knockdown in HepG2 cells leads to an almost complete loss of the ring-like pattern, with a 57.8% reduction in signal intensity relative to scrambled siRNA controls, suggesting a strong dependence of force transmission on N-WASP (Fig. 4o, Supplementary Fig. 10b).

N-WASP is a key activator of Arp2/3-mediated branched actin polymerization, which has been reported to contribute distinctly to myosin-driven versus actin-polymerization-driven behaviors.¹⁴ Our observations are consistent with these prior reports. Moreover, the different effects of N-WASP knockdown in both cell lines closely mirror the results observed upon EIPA treatment. This is consistent with the fact that EIPA inhibits Rac1/Cdc42 activity via suppression of the Na⁺/H⁺ exchanger, and Cdc42 is a key upstream regulator of N-WASP.

Together, these results further demonstrate that the AS1411-nucleolin and Sgc8-PTK7 pairs exhibit distinct dependencies on N-WASP and downstream Arp2/3-mediated branched actin polymerization, which we attribute to fundamentally different force transmission pathways. We added this new data to Fig. 4n-o and Supplementary Fig. 10.

Supplementary Figure 1 a Representative immunostaining images and quantification of N-WASP expression level of HeLa cells after siRNA knockdown. HeLa cells transfected with either scrambled or N-WASP siRNA were stained with anti-N-WASP primary antibody followed by Alexa Fluor™ 647-conjugated secondary antibody. n = 1372 and 1304 cells from 3 replicates for scrambled and siRNA groups. Statistics: unpaired two-tailed Student's *t*-test. **b** Representative immunostaining images and quantification of N-WASP expression level of HepG2 cells after siRNA knockdown. HepG2 cells transfected with either scrambled or N-WASP siRNA were stained with anti-N-WASP primary antibody followed by Alexa Fluor™ 647-conjugated secondary antibody. n = 1222 and 1198 cells from 3 replicates for scrambled and siRNA groups. Statistics: unpaired two-tailed Student's *t*-test. Scale bars = 20 μm.

Fig. 4 n-o Effect of N-WASP on nucleolin- and PTK7-mediated force. **n** Representative brightfield and fluorescence images of HeLa cells after N-WASP siRNA knockdown on AS1411 MP surfaces. Mean fluorescence intensity per cell was quantified. n = 85, 83 cells from 3 replicates for scrambled and siRNA groups. Statistics: unpaired two-tailed Student's *t*-test. **o** Representative brightfield and fluorescence images of HepG2 cells after N-WASP siRNA knockdown on Sgc8 MP surfaces. Mean fluorescence intensity per cell was quantified. n = 45 cells from 3 replicates. Statistics: unpaired two-tailed Student's *t*-test. Scale bar = 20 μm.

A discussion for this result has been added to the main text:

To further distinguish between two force transmission mechanisms, we knocked down the orthogonal regulator neuronal Wiskott–Aldrich syndrome protein (N-WASP). As a critical cytoskeletal regulator, N-WASP interacts with Arp2/3 complex to drive branched actin polymerization,⁵² thereby regulating essential cellular processes such as membrane ruffling, endocytosis, and cytoskeletal remodeling. Inhibition of the Arp2/3 complex has been shown to inhibit Arp2/3-mediated actin polymerization in phagocytic adhesion rings of macrophages but have little effect on integrin-mediated tensions in FAs.⁵³ The latter depends on myosin II-loaded stress fibers which lack Arp2/3 complex. For the AS1411-nucleolin pair, compared with cells transfected with scrambled siRNA, N-WASP knockdown HeLa cells still exhibit a stripe-like force pattern, although the signal amplitude is reduced by 27.9% (Fig. 4n, Supplementary Fig. 10a). This result indicates that force transmission through the AS1411-nucleolin pair is weakly dependent on N-WASP. In contrast, for the Sgc8-PTK7 pair, N-WASP knockdown in HepG2 cells leads to an almost complete loss of the ring-like pattern, with a 57.8% reduction in signal intensity relative to scrambled controls, suggesting a strong dependence on N-WASP-mediated actin dynamics (Fig. 4o, Supplementary Fig. 10b). Moreover, the different effects of N-WASP knockdown in both cell lines closely mirror the results observed upon EIPA treatment (Fig. 4b,h). This is consistent with the fact that EIPA inhibits Rac1/Cdc42 activity, and Cdc42 is a key upstream activator of N-WASP.⁵⁴ Together, these results demonstrate that the AS1411-nucleolin and Sgc8-PTK7 pairs exhibit distinct dependencies on N-WASP and downstream Arp2/3-mediated branched actin polymerization, reflecting fundamentally different force transmission pathways.

Comment 2-3. A critical control is missing to validate the claim that the aptamer-based force sensing is functionally independent of the RGD-mediated adhesion used throughout the study. To robustly prove this

independence, the assay should also be performed on a surface functionalized with the aptamer MPs alone (i.e., without co-immobilized RGD). This would directly test whether baseline integrin adhesion is a prerequisite for the aptamer probes to function, making the conclusion in Figure 2c more convincing.

Response: Thank you. We prepared surfaces functionalized with AS1411 MPs or Sgc8 MPs in the absence of co-immobilized RGD to directly compare unzipping- and shearing-mode of two probes under minimal adhesion conditions. As shown in Fig. 3f, HeLa cells are largely unable to spread on surfaces presenting AS1411 MPs alone, regardless of whether unzipping or shearing probes are used. Unlike the RGD-integrin, the AS1411-nucleolin interaction by itself is insufficient to promote cell adhesion. Compared with conditions in which RGD is co-immobilized, the AS1411 unzipping-mediated mechanosignals are significantly reduced (Fig. 3g), indicating that baseline cell adhesion is a prerequisite for AS1411-based force sensing. We do observe a small fraction of cells that adhered to the surface via nonspecific interactions; notably, these cells are able to activate AS1411 unzipping MPs, further supporting the requirement for baseline adhesion.

Interestingly, Sgc8 MP-functionalized surfaces exhibit a different behavior (Fig.3h). While Sgc8 MPs alone also fail to promote HepG2 cell spreading, the mechanoactivation remains robust (Fig.3i). Even in a non-spread, quasi-suspended state, HepG2 cells effectively activate Sgc8 unzipping MPs, yielding ring-like mechanosignals with intensities comparable to those observed on RGD co-immobilized surfaces. This indicates that for Sgc8-PTK7, baseline cell adhesion is not a prerequisite for mechanoactivation.

Together, the differential dependence of these two aptamer MPs on baseline cell adhesion further supports the existence of distinct force transmission mechanisms. For the AS1411-nucleolin pair, force generation driven by myosin contractility requires mature cell adhesion and myosin-loaded stress fibers. In contrast, for the Sgc8-PTK7 pair, as further elucidated in the mechanistic analysis, force generation arises primarily from membrane ruffling/actin polymerization, with much less dependence on cell adhesion. We have therefore softened the claim of functional independence and added this new data to Fig. 3f-i.

Fig. 3 f Representative brightfield and fluorescence images of HeLa cells cultured on surfaces functionalized with AS1411 MPs alone featuring different force geometries. No co-immobilized RGD-biotin is present to support baseline cell adhesion. Most HeLa cells remain non-adherent and fail to activate the unzipping AS1411 MPs. Only few cells adhering to the surface via nonspecific interactions can activate MPs. **g** Quantification of mean fluorescence intensity per cell for HeLa cells on each geometry. $n = 68, 67$ cells from 3 replicates. Statistics: Two-tailed, Mann-Whitney test. **h** Representative brightfield and fluorescence images of HepG2 cells cultured on surfaces functionalized with Sgc8 MPs alone featuring different force geometries. No co-immobilized RGD-biotin is present to support baseline cell adhesion. Non-spread HepG2 cells activate Sgc8 MPs. **i** Quantification of mean fluorescence intensity per cell for HepG2 cells on each geometry. $n = 71, 68$ cells from 3 replicates. Statistics: Two-tailed, Mann-Whitney test. Scale bars = 20 μm

A discussion for this result has been added to the main text:

Since the above results suggest that aptamers cannot promote cell adhesion, it raises the question to what extent baseline adhesion is required for aptamer MPs to function. To address this, we prepared surfaces functionalized solely with aptamer MPs without co-immobilized RGD to investigate mechanosensing under minimal adhesion conditions. HeLa cells are largely unable to spread on surfaces presenting AS1411 MPs alone, regardless of whether unzipping or shearing MPs are used (Fig.f). Compared with RGD co-immobilized conditions, AS1411 unzipping-mediated mechanosignals are significantly reduced to below 24.4 % (Fig.g), implying that baseline cell adhesion is a prerequisite for AS1411-based force sensing.

Considering that we aim to develop receptor-orthogonal aptamer MPs, we further investigated the Sgc8-aptamer as a potential MP for the protein tyrosine kinase 7 (PTK7), using HepG2 cells that overexpress this receptor. Interestingly, Sgc8 MPs-functionalized surfaces exhibit a somewhat different behavior (Fig.h). Even though Sgc8 MPs alone also fail to promote HepG2 cell spreading, robust mechanoactivation occurs (Fig.i). Even in a non-spread, quasi-suspended state (see brightfield images), HepG2 cells effectively activate Sgc8 unzipping MPs, yielding ring-like mechanosignals. This indicates that for Sgc8-PTK7, baseline cell adhesion is not a prerequisite for mechanoactivation. The differential adhesion dependence of these two aptamer MPs suggests that they transmit forces through fundamentally distinct mechanisms, which we will further discuss below.

Comment 2-4. The actual surface density and stoichiometry of the immobilized RGD and MP molecules were not quantified. These parameters are known to significantly influence cell adhesion and force generation. Providing this characterization would greatly enhance the rigor and reproducibility of the platform.

Response: We performed a reported surface density quantification assay that converts raw fluorescence intensity on our surfaces to the density of fluorescent molecules.^{12,15} Briefly, we used supported lipid bilayers (SLBs) prepared with known stoichiometry of fluorescent lipid as fluorescence standards because their dye density is well defined. We determined the relative molecular brightness (*F* factor) between the fluorophores by comparing the fluorescence intensities of labeled oligonucleotides and small unilamellar vesicles (SUVs) in solution. Using this *F* factor together with a fluorescence calibration curve obtained by imaging SLBs of known dye density, we converted measured fluorescence on our samples to surface concentrations of the oligonucleotide probes.

We generated two independent fluorescence calibration curves (using Texas Red™ DHPE and Oregon Green™ 488 DHPE as standards) and calculated the corresponding *F* factors (Supplementary Fig. 1); these are then applied to independently quantify the surface densities of immobilized RGD and MP molecules. However, direct quantification of the pristine c[RGDfK(Biotin)] molecule is still unfeasible due to its lack of a fluorophore. While using a fluorophore-conjugated RGD-biotin variant is an option, the substantial increase in size and steric hindrance could distort the actual RGD density. Thus, we chose FITC biotin (*Mw* 831.01 g/mol) as a proxy to estimate the RGD density, given its similar molecular weight to c[RGDfK(Biotin)] (*Mw* 829.98 g/mol). It should be noted that this approach ignores potential effects arising from differences in charge and hydrophilicity/hydrophobicity. The surface densities of aptamer MPs and RGD molecules are listed in Supplementary Table 2 now. The different MPs exhibit comparable surface densities in the range of 3500-4500 molecules/μm², and the overall stoichiometric ratio of MPs to RGD is approximately 3:1.

Supplementary Figure 1. Quantification of surface density of aptamer MPs and RGD molecules on the surface. **a-c** Standard curves for quantifying density of aptamer MPs. **a** SLB calibration: Intensity of Texas Red DHPE (Texas Red™ 1,2-dihexadecanoyl-sn-glycero-3-phosphoethanolamine, triethylammonium salt) to the number of molecules on the surface. $n = 15$ from 3 replicates. **b-c** F-factor plot estimation using concentrations of TR-DHPE-derived SUVs (**b**) and RRX-labelled oligonucleotide in solution (**c**) to compare the fluorescence intensity with density. The ratio of the calibration curve slopes was used to determine the “F factor” for the labelled oligonucleotide and the SUV samples. $n = 15$ from 3 replicates. **d-f** Standard curves for quantifying density of RGD molecules. Direct quantification of the pristine c[RGDfK(Biotin)] molecule is infeasible due to its lack of a fluorophore. Using a fluorophore-conjugated RGD-biotin variant may distort the actual RGD density due to the increase in size and steric hindrance. Thus, we chose FITC-biotin (Fluorescein-5(6)-biotinamidohexanoylamidopentylthiourea, Mw 831.01 g/mol) as a proxy to estimate the RGD density, given its similar molecular weight to c[RGDfK(Biotin)] (Mw 829.98 g/mol). It should be noted that this approach ignores potential effects arising from differences in charge and hydrophilicity/hydrophobicity. **d** SLB calibration: Intensity of Oregon Green 488 DHPE (Oregon Green™ 488 1,2-Dihexadecanoyl-sn-Glycero-3-Phosphoethanolamine) to the number of molecules on the surface. $n = 12$ from 3 replicates. **e-f** F-factor plot estimation using concentrations of OG-DHPE-derived SUVs (**e**) and FITC-biotin in solution (**f**) to compare the fluorescence intensity with density. The ratio of the calibration curve slopes was used to determine the “F factor” for the labelled oligonucleotide and the SUV samples. $n = 12$ from 3 replicates.

Supplementary Table 2. Surface density of aptamer MPs and RGD molecules. $n = 18$ from 3 replicates.

	MP density (molecule/ μm^2)	RGD-bio density (molecule/ μm^2) ^a
AS1411 unzipping	4329±394	1297±155
Sgc8 unzipping	4480±239	1320±159
MUC1 S2.2 unzipping	4040±216	1322±101
SYL3c unzipping	4072±166	1356±134
AS1411 shearing	4141±269	1254±82
Sgc8 shearing	3918±609	1321±61
RGD unzipping	3495±201	1374±92
RGD shearing	3581±241	1318±155

^a) mimicked by FITC-biotin molecules with similar molecular weight.

A discussion for this result has been added to the main text:

Supported lipid bilayer-based surface density calibration assays reveal that the densities of different aptamer MPs are in the range of 3500-4500 molecules/ μm^2 , and the overall stoichiometric ratio of MPs to RGD is approximately 3:1 (Supplementary Fig. 1, Supplementary Table 2).

Comment 2-5. The SDR module is currently demonstrated with only a single block–activate cycle, which is not sufficient to establish reversibility. It is recommended that the authors present at least two or more consecutive suppression-recovery cycles. In addition, the RNase H-based “timer” shows tunable activation rates but not a clearly defined lag phase. Including earlier and more frequent time points would help demonstrate a controllable delay before signal onset, and a simple kinetic model correlating decoy concentration with delay would further substantiate the claim of programmability.

Response: Thank you for these suggestions.

(i) SDR module.

To further demonstrate the dynamic reversibility, we conducted a secondary suppression-recovery cycle following the initial cycle. Specifically, after adding the first-round activator for 1 h, the medium was replaced with the second-round blocker to suppress further mechanoactivation. After an additional 0.5 h, a second-round activator was introduced (Fig. 6a). Consistent with the first cycle, the system was incubated for 3 hours prior to imaging. In the second suppression-recovery cycle, addition of the blocker again effectively suppresses the mechanosignals, whereas the subsequent addition of the activator restores the force transmission, mirroring the behavior observed in the initial cycle. The signal level in the second-round blocker condition is higher than that in the first-round blocker group, which we attribute to the cumulative mechanoactivation accumulated over the additional activation period. We added this new data to Fig. 6b and Supplementary Fig. 12e, and discussed it in the text.

Fig. 1 a Schematic of switchable mechanosensing using SDRs for two suppression-recovery cycles. In the 1st cycle, HeLa cells were seeded on surfaces grafted with AS1411 MP for 30 min, followed by addition of 200 nM DNA blocker strands to inhibit aptamer-receptor interaction. After another 30 min, 200 nM DNA activator strands were introduced to displace the blockers via toehold-mediated SDR, restoring mechanosensing. Mechanosignals were quantified after 3 h. In the 2nd cycle, another round of blocker strands was introduced 1 h after adding activator strands in the 1st cycle. After an additional 30 min, a second-round activator was introduced. Consistent with the 1st cycle, the system was incubated for 3 h prior to imaging. **b** Quantification of mean fluorescence intensity per cell of HeLa cells on AS1411 MP surfaces after sequential blocker and activator treatment for two suppression-recovery cycles. 1st cycle, $n = 54$ cells from 3 replicates. 2nd cycle, $n = 48$ cells from 3 replicates. Statistics: 1st cycle, One-way ANOVA with Bonferroni post-hoc tests. 2nd cycle, Two-tailed, Mann-Whitney test.

Supplementary Figure 12 e Representative brightfield and fluorescence images of HeLa cells on AS1411 MP surfaces after sequential blocker and activator treatment. Scale bars = 20 μm .

A discussion for this result has been added to the main text:

The second blocker-activator cycle closely mirrors the suppression-recovery behavior observed in the initial cycle (Supplementary Fig. 12e). The overall higher signal level can be attributed to cumulative mechanoactivation accumulated during the additional activation period.

(ii) RNA-RNase H module.

We monitored the reaction kinetic of RNA-RNase H module to assess reaction rates. A FRET pair (Atto488/BHQ1) was incorporated into the AS1411 strand and RNA blocker so that we can quantify the ratio of AS1411 aptamer reconfiguration. We assumed pseudo first-order kinetics and fitted the data using one phase exponential decay function. The reconfiguration of aptamer MPs does not reach saturation by 2 h (the onset of measurable mechanoactivation) due to a relatively low RNase H concentration used (10 U/mL) before. Therefore, no clearly defined lag phase is observed (Supplementary Fig. 13b and Supplementary Table 3). Nevertheless, the introduction of decoy strand exerts a clear, dose-dependent inhibitory effect on the reaction rate of RNA blocker degradation: Consistent with this trend, we monitored mechanoactivation on the cell level at 0.5 h and 1 h after cell seeding and find that the decoy-induced reduction in reaction rate and the extent of MP reconfiguration correlates with the onset of force signals (Fig. 6h, Supplementary Fig. 13c). Compared with 1 nM decoy, 10 nM decoy slightly delays the rate of signal increase, whereas 100 nM decoy markedly prolonged the time required to reach a signal amplitude comparable to that achieved with 1 nM decoy at 2 h, extending it to approximately 7 h.

Supplementary Figure 13. **b** Plot of normalized surface atto488 intensity versus time to compare the reconfiguration rate of AS1411 aptamer in RNA-RNase H module in the presence of decoy strands. Surface-immobilized AS1411 MPs are blocked with RNA blocker and degraded by RNase H at 10 or 100 U/mL and decoy strands at 1-100 nM. Reaction medium: 1 % FBS, 1 % P/S, 100 μ g/mL actin protein, DMEM. $n = 15$ from 3 replicates. At each reaction time point, the surface Atto488 intensity is normalized to that of unblocked AS1411 MPs labeled with atto488, which serves as the positive control. **c** Representative brightfield and fluorescence images of HeLa cells on RNA-blocked AS1411 MP surfaces after adding 10 U/mL RNase H and varying RNA/DNA duplex concentrations. Scale bars = 20 μ m.

Supplementary Table 3. Reaction rate constant of SDR and RNA-RNase H modules. To understand the reconfiguration efficiency and tunability of different DNRs, we quantified the surface reconfiguration kinetics. The data were analyzed assuming pseudo-first-order kinetics and fitted using one phase exponential decay function. For the SDR module, reactivation with 200 nM DNA activator yields a reconfiguration rate of $2.14 \times 10^{-2} \text{ min}^{-1}$. The RNA-RNase H module enables degradation rate tuning by adjusting the initial RNase H concentration, resulting in accelerated (100 U/mL, $5.26 \times 10^{-2} \text{ min}^{-1}$) or decelerated (10 U/mL, $0.71 \times 10^{-2} \text{ min}^{-1}$) reconfiguration kinetics. Moreover, at a fixed RNase H concentration (10 U/mL), the RNA-RNase H module allows the introduction of non-enzymatic RNA/DNA decoys, providing an additional orthogonal layer of regulation.

	$k (\times 10^{-2} \text{ min}^{-1})$	R^2
SDR 200 nM DNA activator	2.14	0.9926
RNase H 100U/mL	5.26	0.9925
RNase H 10U/mL	0.71	0.9842
RNase H 10U/mL+1 nM rdm RNA/DNA decoy	0.73	0.9782
RNase H 10U/mL+10 nM rdm RNA/DNA decoy	0.42	0.9603
RNase H 10U/mL+100 nM rdm RNA/DNA decoy	0.11	0.9831

Fig. 2 h Corresponding quantification of mean fluorescence intensity per cell. Reaction medium: 1 % FBS, 1 % P/S, 100 μ g/mL actin protein, DMEM. n = 27 cells from 3 replicates. Scale bars = 20 μ m.

A discussion for this result has been added to the main text:

The introduction of decoy strand exerts a clear, dose-dependent inhibitory effect on the reaction rate of RNA blocker degradation (Supplementary Fig. 13b, Supplementary Table 3). Consistent with this trend, 10 nM decoy is sufficient to slightly delay the mechanoresponse, whereas 100 nM nearly abolishes the response within 4 h and a significant response is observed only at 7 h (Fig. 6g,h, Supplementary Fig. 13c).

Comment 2-6. The statement that aptamer MPs exhibit “enhanced force selectivity” compared to RGD MPs may be misleading. A more accurate interpretation is that AS1411–nucleolin interactions occur within a lower physiological force regime (consistently below 54 pN), while RGD–integrin forces readily exceed this threshold. This distinction reflects biological force domains rather than an intrinsic property of the probe.

Response: Thanks for bringing this up. We have revised the subsection title and re-written the corresponding text to make it clearer.

Comment 2-7. Regarding the citation of Yang et al. (Ref. 20), the manuscript currently describes this study as “force being sensed using a DNA MP”, which may not fully reflect its contribution. Specifically, Yang et al. provided the proof-of-concept that a DNA aptamer, taking CI-M6RP-targeting aptamer as the example, can function as a force-reception module to sense weak endocytic forces with fluorescent signal readout and then translate these force cues into intracellular signals via DNA dynamic reactions. Thus, a more balanced description should acknowledge this advance in the revision, and it is recommended to add a brief discussion clarifying how the present work extends this concept.

Response: We thank the reviewer for pointing this out and have properly credited this work in the Introduction Section:

Indeed, a recent proof-of-concept using the CI-M6PR-targeting aptamer demonstrated that an aptamer can function as a force-reception module to sense weak endocytic forces (~ 4 pN)²⁴ and then translate this force cue into intracellular signals via DNA-mediated receptor dimerization.²⁰

We included a comparison in the Discussion Section:

These newly uncovered mechanisms also expand the potential of aptamers as versatile force-reception modules for bidirectional mechano-biochemical signal transduction. Beyond relying on weak DNA hairpins to

sense clathrin-mediated endocytic forces (~ 4 pN),²⁰ cells can exert forces >12 pN on aptamer MPs. Such forces allow the irreversible exposure of cryptic ssDNA domains that can trigger downstream reactions such as transcription, hairpin chain reactions,²¹ or rolling circle amplification.⁵⁹ This expanded force range substantially broadens the design space for intelligent, force-responsive material interfaces capable of cellular adaptation.

Minors

Comment 2-8. Use more precise terminology such as “receptor-low” or “low-expressing” instead of “receptor-negative,” as the supplementary flow cytometry data indicate non-zero expression.

Response: Thank you. We have replaced the terminology “receptor-negative” and “receptor-positive” with “low-expressing” and “high-expressing,” respectively.

Comment 2-9. Clarify in the text or figure legend for Figure 3 why both “mean” and “integrated” intensity are presented, and how each metric provides distinct insights.

Response: Thank you. Please see **Response 1-4** above.

A discussion for this result has been added to the main text:

The integrated intensity reflects the cumulative mechanosignal across the entire cell footprint and is thus not only regulated by intrinsic T_{tot} of the probe but inherently dependent on cell area.¹⁹

We also updated the figure captions:

Fig.3 c-e Quantification of (c) mean fluorescence intensity per cell, (d) integrated fluorescence intensity per cell, and (e) cell spreading area for HeLa cells on each MP type and geometry. $n = 49, 47, 45, 36$ (left to right) cells from 3 replicates. Statistics: Kruskal-Wallis test with Dunn’s multiple comparisons. Note that both mean and integrated intensity are shown because mean intensity mainly captures the T_{tot} -dependent shifts while integrated intensity is cross-regulated by both the intrinsic T_{tot} of the probe and the cell spreading area.

Fig.5 b Corresponding quantification of mean and integrated fluorescence intensity per cell of HeLa and HepG2 cells on MUC1 S2.2 MP surfaces. $n = 31, 27$ cells from 3 replicates. Statistics: Two-tailed, Mann-Whitney test. Note that both mean and integrated intensity are shown here because different cell lines may have distinct basal morphologies with different cell area.

Comment 2-10. In Figure 6, the restored force signal distribution appears inconsistent across conditions (e.g., recovery mainly at the cell edge vs. both edge and center). The authors should clarify whether these differences reflect mechanistic variations or are attributable to experimental variability.

Response: Thank you for the question. We have carefully checked the spatial distribution of the force signals in HeLa cells. The characteristic force pattern mediated by the AS1411-nucleolin interaction is defined by strong stripe-like signals at the cell edge, like focal adhesions, whereas signals in the central region are consistently weaker. We examined all images showing restored force signals. Under comparable restoration conditions, the overall force patterns are consistent. For example, the force distributions observed in Fig. 6d (RNase H, 10 U/mL, 4 h) and Fig. 6g (RNase H, 10 U/mL+ 1 nM decoy, 4 h) are largely similar. We do acknowledge that the intensity of the central signal can vary between individual cells. For instance, in Fig. 6g, the 1 nM decoy group (7h) exhibits central signals below the average, while the 10 nM decoy group (7h) shows relatively higher central intensity. Despite these fluctuations, the fundamental spatial logic where peripheral stripe-like signals dominate over central signals remains conserved across most of the groups. We therefore attribute these differences to individual cellular heterogeneity and experimental variability rather than a shift in the

mechanisms.

Comment 2-11. For PTK7-mediated forces, it remains unclear whether the signal reflects protrusive pushing or contractile pulling. The schematic suggests pulling, whereas the text emphasizes polymerization-driven pushing. This ambiguity should be explicitly acknowledged and discussed.

Response: Thank you for pointing this out, and we apologize for this mistake in the schematic. Based on our previous data and the new analyses presented in Responses 2-2 and 2-3, PTK7-mediated forces arise predominantly from actin polymerization-driven pushing. We have corrected the schematic accordingly now. All related discussions have been revised to consistently emphasize this actin polymerization-driven pushing.

Comment 2-12. In the discussion, briefly acknowledge the limitations of DRN-controlled systems for potential in vivo applications, including delivery challenges, nuclease stability, and cytotoxicity.

Response: Thank you. We have added the related discussion in the text:

However, long-term application in these environments still faces stability challenges. While modifications like peptide nucleic acids,⁶³ phosphorothioate linkages,⁶⁴ or our recent L-DNA strategy protect the MP structure,⁶⁵ single-stranded aptamers remain inherently degradation-prone. Balancing their structural integrity with binding affinity is critical. In this context, using nuclease inhibitors such as actin or decoy DNA may be more effective.^{66,67} Additionally, for potential in vivo applications, deep-tissue diffusion, off-target effects arising from protein corona formation, and potential immunotoxicity remain significant challenges.

Comment 2-13. While the manuscript identifies novel force pathways, the discussion could be enriched by addressing their biological significance. Why do cells exert forces through these non-canonical receptors? Furthermore, since detailed analysis was limited to nucleolin and PTK7, it would be valuable to comment on the potential generalizability to other possible aptamer-receptor pairs with biological functions.

Response: Thank you. The transient coupling between these non-canonical receptors and the cytoskeleton provides the fundamental structural basis for force transmission, while their biological functions are still unknown at this stage and should not become subject to excessive speculation in this manuscript. Endocytic force is a classical form of actively applied cellular force acting on membrane receptors and is associated with well-defined biological functions. During endocytosis, receptors are coupled to endocytic machinery, such as clathrin and the cytoskeleton, generating forces of ~4.7 pN. Through this canonical process, cells actively internalize and recycle membrane proteins to maintain signaling homeostasis.

In our study, however, the unzipping probes selectively filter out forces below 12 pN, indicating that the forces we observe are unlikely to arise from receptor internalization or recycling. Instead, the relatively high forces associated with nucleolin or PTK7 likely originate from their coupling to more mechanically active, high-force cytoskeletal structures. Whether this force transmission reflects an active, regulated cellular process or a passive mechanical linkage remains unclear, and to our knowledge, there is currently no direct evidence demonstrating a definitive function associated with such forces. This highlights the importance of revisiting the biological functions of both these aptamers and receptors from a mechanobiological perspective. For example, whether AS1411-mediated processes such as angiogenesis are modulated by mechanical forces remains an open question. Although we are highly interested in these directions, they extend beyond the primary scope of the present work. Notably, in our experiments, unlike mobile native ligands, aptamer MPs are stably immobilized on the surface, thereby providing sufficient reaction force to support force transmission. It is also of significant interest to explore whether cryptic motifs within the native extracellular matrix can

interact with nucleolin with similar force transmission, and regulate cellular homeostasis through non-canonical mechanotransduction pathways. A prime candidate for such an interaction is the precursor of the nucleolin ligand endostatin, located in the NC1 domain of Type XVIII collagen,⁸ which may serve as a native mechanical anchor despite potential differences in binding affinity.

Finally, as suggested by the reviewer, the generalizability of the aptamer-based platform provides opportunities to probe more established aptamer–receptor pairs with known biological functions, such as PD-L1-targeting aptamers involved in immune checkpoint regulation,^{16,17} or CD28 costimulatory aptamers in T cells,¹⁸ to determine whether their biological activities are also regulated by mechanical force. Exploring these questions represents an important and promising direction for future research.

We have revised the related discussion in the text:

The identification of noncanonical forces highlight the importance of revisiting the biological functions of both these aptamers and receptors from a mechanobiological perspective. Determining whether these receptors are directly coupled to classical adhesion structures or are independently linked to the same actomyosin network via intermediary components will be essential for uncovering more general principles. Whether similar interactions occur under native conditions remains to be elucidated. A prime candidate is the precursor of the nucleolin ligand endostatin, located in the NC1 domain of type XVIII collagen,⁶⁰ which may serve as a native anchor for nucleolin despite potential differences in binding affinity. Addressing these questions could reveal how cryptic motifs within the ECM regulate cellular homeostasis through non-canonical mechanotransduction pathways. Meanwhile, caution is required when extrapolating from aptamer-receptor pairs to native ligand-receptor interaction, particularly regarding the immobilization state of the ligand, binding affinity and bond mechanics, and the multivalency and cooperative effects inherent to native ligands. For example, the AS1411 aptamer binds nucleolin with a K_d of 69.1 nM,²⁸ which is lower than that of its native ligand endostatin (23.2 nM),⁶¹ suggesting that our measurements may slightly underestimate the native capacity for tension generation. Finally, the generalizability of the aptamer-based platform provides opportunities to probe other established aptamer-receptor pairs with well-defined biological functions, such as PD-L1-targeting aptamers involved in immune checkpoint regulation,^{62,63} or CD28 costimulatory aptamers in T cells,⁶⁴ to determine whether their biological activities are similarly regulated by mechanical force.

References

1. Jo, M. H. *et al.* Single-molecule characterization of subtype-specific $\beta 1$ integrin mechanics. *Nat. Commun.* **13**, 7471 (2022).
2. Song, N. *et al.* The nuclear translocation of endostatin is mediated by its receptor nucleolin in endothelial cells. *Angiogenesis* **15**, 697–711 (2012).
3. Shi, H. *et al.* Nucleolin is a receptor that mediates antiangiogenic and antitumor activity of endostatin. *Blood* **110**, 2899–2906 (2007).
4. Koutsoumpa, M. *et al.* Interplay between $\alpha v\beta 3$ integrin and nucleolin regulates human endothelial and glioma cell migration. *J. Biol. Chem.* **288**, 343–354 (2013).
5. Bi, J. *et al.* Identification of nucleolin as a lipid-raft-dependent $\beta 1$ -integrin-interacting protein in A375 cell migration. *Mol. Cells* **36**, 507–517 (2013).
6. Ding, Y. *et al.* Heat shock cognate 70 regulates the translocation and angiogenic function of nucleolin. *Arterioscler. Thromb. Vasc. Biol.* **32**, e126–e134 (2012).
7. Huang, Y. *et al.* The angiogenic function of nucleolin is mediated by vascular endothelial growth factor and nonmuscle myosin. *Blood* **107**, 3564–3571 (2006).
8. Heljasvaara, R. *et al.* Generation of biologically active endostatin fragments from human collagen XVIII by distinct matrix metalloproteases. *Exp. Cell Res.* **307**, 292–304 (2005).
9. Ma, R. *et al.* Molecular mechanocytometry using tension-activated cell tagging. *Nat Methods* **20**, 1666–1671 (2023).
10. Rashid, S. A. *et al.* DNA Tension Probes Show that Cardiomyocyte Maturation Is Sensitive to the Piconewton Traction Forces Transmitted by Integrins. *ACS Nano* **16**, 5335–5348 (2022).
11. Pawlak, M. R. *et al.* RAD-TGTs: high-throughput measurement of cellular mechanotype via rupture and delivery of DNA tension probes. *Nat. Commun.* **14**, 2468 (2023).
12. Velusamy, A., Sharma, R., Rashid, S. A., Ogasawara, H. & Salaita, K. DNA mechanocapsules for programmable piconewton responsive drug delivery. *Nat. Commun.* **15**, 704 (2024).
13. Peng, Y.-H. *et al.* Dynamic matrices with DNA-encoded viscoelasticity for cell and organoid culture. *Nat. Nanotechnol.* **18**, 1463–1473 (2023).
14. Kundu, S., Pal, K., Pyne, A. & Wang, X. Force-bearing phagocytic adhesion rings mediate the phagocytosis of surface-bound particles. *Nat. Commun.* **16**, 984 (2025).
15. Duan, Y. *et al.* Mechanically triggered hybridization chain reaction. *Angew. Chem. Int. Ed.* **60**, 19974–19981 (2021).
16. Wang, D. *et al.* Enrichment and sensing tumor cells by embedded immunomodulatory DNA hydrogel to inhibit postoperative tumor recurrence. *Nat. Commun.* **14**, 4511 (2023).
17. Li, Y. *et al.* Covalent LYTAC enabled by DNA aptamers for immune checkpoint degradation therapy. *J. Am. Chem. Soc.* **145**, 24506–24521 (2023).
18. Zhou, J. & Rossi, J. Aptamers as targeted therapeutics: current potential and challenges. *Nat. Rev. Drug. Discov.* **16**, 181–202 (2017).